# Drug reformulation for a neglected disease. The NANOHAT project to develop a safer more effective sleeping sickness drug

Lisa Sanderson[1]☉, Marcelo da Silva[1]☉, Gayathri N. Sekhar[1], Rachel C. Brown[1], Hollie Burrell-Saward[2], Mehmet Fidanboylu[1], Bo Liu[3], Lea Ann Dailey[1¤], Cécile A. Dreiss[1], Chris Lorenz[4], Mark Christie[1], Shanta J. Persaud[3], Vanessa Yardley[2], Simon L. Croft[2], Margarita Valero[5], Sarah A. Thomas[1]*

**1** King's College London, Institute of Pharmaceutical Science, Franklin-Wilkins Building, Stamford Street, London, United Kingdom, **2** Faculty of Infectious and Tropical Diseases, London School of Hygiene and Tropical Medicine, London, United Kingdom, **3** King's College London, Department of Diabetes, School of Life Course Sciences, Faculty of Life Sciences & Medicine, London, United Kingdom, **4** King's College London, Theory & Simulation of Condensed Matter Group, Department of Physics, Strand, London, United Kingdom, **5** Physical Chemistry Department, Faculty of Pharmacy, University of Salamanca, Salamanca, Spain

☉ These authors contributed equally to this work.
¤ Current address: Dept. of Pharmaceutical Technology and Biopharmacy, University of Vienna, Vienna, Austria
* sarah.thomas@kcl.ac.uk

**Data Availability Statement:** All relevant data are within the manuscript and its supporting Information files.

## Abstract

### Background

Human African trypanosomiasis (HAT or sleeping sickness) is caused by the parasite *Trypanosoma brucei sspp*. The disease has two stages, a haemolymphatic stage after the bite of an infected tsetse fly, followed by a central nervous system stage where the parasite penetrates the brain, causing death if untreated. Treatment is stage-specific, due to the blood-brain barrier, with less toxic drugs such as pentamidine used to treat stage 1. The objective of our research programme was to develop an intravenous formulation of pentamidine which increases CNS exposure by some 10–100 fold, leading to efficacy against a model of stage 2 HAT. This target candidate profile is in line with drugs for neglected diseases inititative recommendations.

### Methodology

To do this, we evaluated the physicochemical and structural characteristics of formulations of pentamidine with Pluronic micelles (triblock-copolymers of polyethylene-oxide and polypropylene oxide), selected candidates for efficacy and toxicity evaluation *in vitro*, quantified pentamidine CNS delivery of a sub-set of formulations *in vitro and in vivo*, and progressed one pentamidine-Pluronic formulation for further evaluation using an *in vivo* single dose brain penetration study.

### Principal Findings

Screening pentamidine against 40 CNS targets did not reveal any major neurotoxicity concerns, however, pentamidine had a high affinity for the imidazoline$_2$ receptor. The reduction

**Funding:** This research was funded by a Medical Research Council (MRC) developmental pathway funding scheme (DPFS) award [MR/K015451/1] to SAT (PI), CAD, CL, SJP, MC (all at King's College London), VY and SLC (both at LSHTM) (https://mrc.ukri.org/funding/browse/biomedical-catalyst-dpfs/biomedical-catalyst-developmental-pathway-funding-scheme-dpfs-submission-deadlines/ accessed 20.03.2020). This research was also supported by a MRC PhD studentship [MR/K500811/1] (to GNS and supervised by SAT and CAD; https://mrc.ukri.org/ accessed 20.3.2020). The BBSRC Centre of Integrative Biomedicine provided funding for the haemolysis assay (SAT and LAD) [BB/E527098/1] (https://bbsrc.ukri.org/ accessed 20.3.2020). A multi-user equipment grant from the Wellcome Trust provided a Perkin-Elmer Tricarb 2900TR liquid scintillation counter [080268] to SAT (PI) (https://wellcome.ac.uk/ accessed 20.3.2020). The funders had no role in study design, data collection and analysis, decision to publish, or preparation of the manuscript.

**Competing interests:** The authors have declared that no competing interests exist.

in insulin secretion in MIN6 β-cells by pentamidine may be secondary to pentamidine-mediated activation of β-cell imidazoline receptors and impairment of cell viability. Pluronic F68 (0.01%w/v)-pentamidine formulation had a similar inhibitory effect on insulin secretion as pentamidine alone and an additive trypanocidal effect *in vitro*. However, all Pluronics tested (P85, P105 and F68) did not significantly enhance brain exposure of pentamidine.

## Significance

These results are relevant to further developing block-copolymers as nanocarriers, improving BBB drug penetration and understanding the side effects of pentamidine.

## Author summary

Sleeping sickness or human African Trypanosomiasis (HAT) is a disease caused by a parasite, which is transferred to humans by the bite of an infected tsetse fly. There are two disease stages: the first stage is the blood-based stage of the disease and the second stage affects the brain. It is fatal if left untreated. The blood-brain barrier (BBB) makes the brain stage difficult to treat because it prevents 99% of all drugs from entering the brain from the blood. Those anti-HAT drugs that do enter the brain are toxic and have serious side effects. Pentamidine is a less toxic blood stage drug, which our research has shown has a limited ability to cross the BBB due to its removal by proteins called transporters. The objective of this study was to use Pluronic to improve pentamidine delivery to target sites, whilst reducing its side effects. Pluronic is a polymer, which can assemble into micelles and encapsulate the drug. Thus, prolonging its circulation time and protecting it. Our study indicated that the selected Pluronics did not increase the brain delivery of pentamidine. However. Pluronic-pentamidine formulations were identified that harboured trypanocidal activity and did not increase safety concerns compared to unformulated pentamidine.

## Introduction

Human African trypanosomiasis (HAT or sleeping sickness) is a potentially fatal disease caused by the parasite *Trypanosoma brucei sspp*. Recent epidemiological studies in 30 of the 36 African countries listed as endemic for the disease indicate that, whilst the number of disease cases has been decreasing since 1990, there are still ~4,000 new infections/year, and ~15,000 cases worldwide [1, 2]. Furthermore, there is a substantial unreported burden of HAT [3].

The disease has two stages–a haemolymphatic stage after the bite of an infected tsetse fly, followed by a central nervous system (CNS) stage when the parasite penetrates the brain, causing death if left untreated. The blood-brain barrier (BBB) makes the CNS stage difficult to treat because it prevents 99% of all known compounds from entering the brain, including most anti-HAT drugs [4–7]. Those that do enter the brain are toxic compounds, can have serious side effects, are complex to administer and/or are expensive. Pentamidine is a less toxic blood stage drug, which is known to treat early-late (transition) stage HAT[8], but cannot treat stage 2 disease as it does not sufficiently penetrate the BBB[7] and it causes peripheral side effects (e.g. hypoglycaemia (incidence 5–40%) and diabetes mellitus (incidence: occasional but irreversible)[9] which preclude increasing the dose to overcome this limitation. Research has shown pentamidine has a limited ability to cross the BBB and reach the brain due to it

**Table 1. Pluronics used in this Study, with their Name, Block Composition, Hydrophilic-Lipophilic Balance (HLB) and General Formula.** L, F or P refers to Liquid, Flake, or Paste Physical Forms, respectively.

| Poloxamer | Pluronic | MW | Number of EO blocks | Number of PO blocks | HLB | Formula |
|---|---|---|---|---|---|---|
| 235 | P85 | 4600 | 52.27 | 39.66 | 16 | $EO_{26.13}PO_{39.66}EO_{26.13}$ |
| 335 | P105 | 6500 | 73.86 | 56.03 | 15 | $EO_{36.93}PO_{56.03}EO_{36.93}$ |
| 188 | F68 | 8400 | 152.73 | 28.97 | 29 | $EO_{76.37}PO_{28.97}EO_{76.37}$ |
| 181 | L61 | 1950 | 4.55 | 31.03 | 3 | $EO_2PO_{30}EO_2$ |

physicochemical characteristics and its removal by the efflux transporters P-glycoprotein (Pgp) and multi-drug resistance associated protein (MRP) [7] (Fig A in S1 File). Furthermore, transporters are considered essential in the mode of action of pentamidine against trypanosomes.

Poloxamers, with commercial trademark Pluronics (BASF) or Synperonics (CRODA), are triblock copolymers made of two poly(ethylene oxide) (PEO) blocks interspaced by a poly(pro-pylene oxide) (PPO) block and follow the general basic formula: $PEO_x-PPO_y-PEO_x$, where x and y are the size of PEO and PPO blocks, respectively (Table 1). In an aqueous environment and above the critical micelle concentration (CMC), the copolymers self-assemble into micelles, with the PEO chains forming a hydrophilic shell around a PPO hydrophobic core, within which lipophilic drugs can be solubilised, drug-free fraction decreased and circulation time increased [10]. A variety of Pluronic block copolymers differing in the lengths of the EO and PO blocks are available for formulation with pharmaceutical drugs. Importantly the size of the hydrophobic block affects micellization and drug solubilisation[11]. Furthermore, com-bining different Pluronics can enhance drug/micelle interactions and drug loading[12, 13]. The PEO shell serves as a stabilizing layer between the hydrophobic core and the external medium, and prevents aggregation, plasma protein adsorption and opsonization and therefore recognition by the macrophages of the reticuloendothelial system [14]. Pluronic copolymers are also endowed with low cytotoxicity and weak immunogenicity in topical and systemic administration. Even though PEO–PPO–PEO materials are non-degradable, molecules with a molecular weight (MW) <7 kDa can be filtered by the kidney and cleared in urine[15] (Table 1). In addition, Pluronics are recognised pharmaceutical excipients listed in the US and British Pharmacopoeia so have an established safety profile.

Thus Pluronics have attracted a great deal of attention in pharmaceutical applications as drug solubilisers [14] or controlled drug-release agents [13, 16, 17]. Notably, Pluronic P85, P105, F68 and L61 have been shown to inhibit efflux transporters (including P-gp and MRP1-2) and have been shown to enhance drug passage across the BBB [16, 18–29]. They have all been approved as cosmetic ingredients [15] with F68 having been utilized as a blood substitute component[30]. Transporter-targeting Pluronics (L61 and F127) have successfully completed a phase 2 clinical trial for the intravenous treatment of adenocarcinoma of the upper gastroin-testinal tract [31, 32]. Interestingly, F127-based amphotericin B-containing micelles have been shown to be highly effective in treating *Leishmania amazonensis*-infected BALB/c mice with results indicating that the empty micelles also exhibited antileishmanial activity [33]. Together these studies demonstrate that Pluronics have potential beyond the traditional role of simple micellar vessels for drug encapsulation and longer circulation, but are also active agents with key biological functions [34].

In this Medical Research Council (MRC) developmental pathway funding scheme (DPFS) study our multi-disciplinary team developed a milestone driven progression strategy (Fig 1) in order to assess the potential of pentamidine-Pluronic formulations to effectively treat stage 2 disease, reduce the major known side effect of pentamidine on the pancreas and shorten the

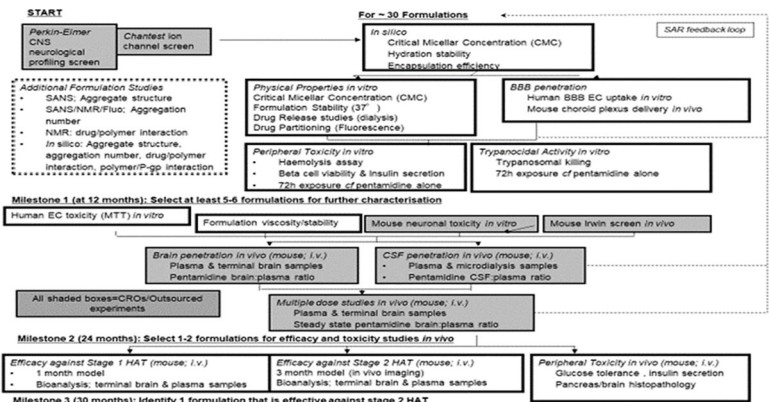

**Fig 1. NANOHAT project screening cascade.** We used a structure activity relationship (SAR) feedback loop to further refine the selection of the lead formulations progressing through the screening cascade We screened approximately 30 pentamidine/Pluronic formulations during this project using this rational, iterative approach. The three milestones were intended to ensure that the most appropriate formulations, on the basis of *in silico* and *in vitro* data, were taken forward to the *in vivo* pharmacokinetic studies and that the formulations with the greatest likelihood of success would be tested in the whole animal efficacy studies as outlined in the progression strategy.

length of treatment required to treat stage 1 disease. It was anticipated that the benefits of this approach would be a combined pentamidine-Pluronic formulation which would provide a single therapeutic entity for safer, simpler and more cost-effective treatment of all HAT stages using an established drug with a known safety profile. Four Pluronics were selected for evaluation based on their block-copolymer architecture, established safety profile and known ability to inhibit Pgp. These were P85, P105, F68 and L61 (Table 1). An iterative approach was utilized as illustrated in Fig 1

## Methods

### Ethics statement

All animal studies were performed within the framework of the Animals Scientific Procedures Act (1986) and Amendment Regulations 2012 and with consideration to the ARRIVE guidelines. The study was approved by the King's College London Animal Welfare and Ethical Review Body or the London School of Hygiene and Tropical Medicine Ethics Committee and Animal Welfare and Experimental Research Board, as appropriate.

### Materials

Pentamidine (1,5-bis-4ρ-amidinophenoxypentane) isethionate salt (MW 592.68; 98% purity; catalogue number P0547) and Pluronic P105 (batch number BCBP8915V) were purchased from Sigma Aldrich (Poole, Dorset, UK). Pluronic P85 (mat 30085877 batch number: WPYE5378) was a kind donation from BASF plc (Cheshire, UK). Pluronic F68 (medical grade Catalogue number 2750016; batch numbers M7102 and MR29468) was purchased from MP Biomedicals, LLC (Illkirch Cedex, France). L61 was purchased from Aldrich (catalogue number 435422; batch number MKBH8737V). Purity of excipients met US Pharmacopeia convention NF32 specifications and was confirmed by external specialist laboratory (Text A in S1 File).

### Evaluation of potential neurotoxicity of pentamidine

New toxicities may arise following pentamidine's improved access to the CNS. The potential of pentamidine to cause neurotoxicity was evaluated by a brief review of the literature together

with a neurological profiling screen and ion channel activity screens. The biological screens were performed by external specialist laboratories as described below.

**Neurological profiling screen.** A CNS side effect panel was custom designed and binding assays performed by Perkin-Elmer Science Discovery Systems (Hanover MD 21076, USA). The $IC_{50}$ for pentamidine against *Trypanosoma brucei brucei* strain 427 and *Trypanosoma brucei gambience* has been reported as 1.8–26.1 nM and 14.7±4.7 nM respectively [35–37]. Thus, testing was performed at a single concentration of 1 μM (100-times the trypanocidal concentration), with follow up concentration-response curves in any assay where there was greater than 70% inhibition to determine an inhibition constant ($K_i$).

**Ion channel (hKir2.1) activity screens.** The *in vitro* effects of pentamidine isethionate on cloned hKir2.1 potassium channels (encoded by the human KCNJ2 gene) responsible for the $I_{K1}$, inwardly rectifying potassium current, were examined by ChantTest Corporation (Cleveland Ohio 44128, USA) to industry standards (Chantest FastPatch Assay; study number. 130827. DCC). Human epithelial kidney 293 (HEK293) cells (ATCC, Manassas VA USA) were stably transfected with the appropriate ion channel cDNA encoding the pore-forming channel unit. Cells were cultured in Dulbecco's Modified Eagle Medium / Nutrient Mixture F-12 (D-MEM/F-12) supplemented with 10% foetal bovine serum, 100 U/mL penicillin G sodium, 100 μg/mL streptomycin sulphate and 500 μg/mL G418. Cultured cells were maintained in a tissue culture incubator set at 37˚C in a humidified 95% air and 5% $CO_2$ atmosphere. Pentamidine was dissolved in HEPES-buffered physiological saline containing 0.3% DMSO and sonicated (Model 2510/5510, Branson Ultrasonics, Danbury, CT) at room temperature for at least 20 minutes. A glass-lined 96 well compound plate was loaded with the appropriate amount of test (five different concentrations) and positive control (100μM $BaCl_2$) solutions, and placed in the plate well of the QPatchHT (Sophion Bioscience A/S, Denmark). All experiments were performed at room temperature. Each cell acted as its own control. Vehicle was applied to naïve cells for a 5–10 minute exposure interval. The test solution applied for a minimum of three minutes via the QPatch robot pipetting system to naïve cells (n≥2, where n = the number of cells/concentration). Each solution exchange on the QPatch, performed in quadruplicate, consisted of a 5 μl exchange through the microfluidic flow channel, resulting in 100% replacement of the compound in the QPlate. Intracellular solution was loaded into the intracellular compartments of the QPlate planar electrode (130mM K-Asp, 5mM $MgCl_2$, 5 mM EGTA, 4mM Tris-ATP and 10 mM HEPES). Cell suspension was pipetted into the extracellular compartments of the QPlate planar electrode.

Onset and steady state block of hKir2.1 current was measured using a ramp protocol with fixed amplitudes (hyperpolarization: -110 mV, 200 ms duration, followed by a 1-second ramp from -110 mV to +50 mV) repeated at 10 s intervals from a holding potential of –70 mV. Current amplitude was measured at the end of the step to -110 mV. Leak current was calculated and subtracted from the total membrane current record.

## Determination of the micellar aggregation properties of Pluronic

The CMC, micellar size and aggregation number were determined in different solvents, using a unique combination of light and neutron scattering and atomistic simulations. We also measured the partitioning of pentamidine isethionate in selected Pluronic and the *in vitro* release profile.

**Preparation of solutions for physicochemical measurements.** Unless stated, F68, P85, P105 or L61 were either dissolved in water (aqueous) or saline solution (0.9% w/v sodium chloride solution). Pluronic mixtures were also prepared either with a fixed mass ratio of 1:1 (F68-P105 or F68-P85) or in the case of L61, 0.01%. Samples were left to equilibrate for at least 3 hours prior to any measurement. Ultra-pure water (18.2 MΩ·cm—Millipore-filtered) was used throughout the experiments.

**Phase behaviour.**   In this study, L61 alone and in mixtures with one or two other Pluronics in both water (aqueous) and saline mediums were visually assessed from 20˚C to 50˚C in 5˚C steps, plus 37˚C, to assess the impact of mixtures on L61 cloud point (24˚C for a 1% solution) [38].

**CMC determination by fluorescence spectroscopy.**   The CMC determines thermodynamic stability of the micelles during dilution of the drug delivery system in body fluids [11, 17]. Furthermore, CMC is an important parameter in view of the biological response modifying effects of Pluronic block copolymers since it is needed to determine the maximum achievable concentration of the polymer single chains ("unimers") [21]. For measurement of the CMC, pyrene (Sigma catalogue number 82648; pyrene puriss p.a. for fluorescence, $\geq$99%) was used as a probe. A stock solution of pyrene in acetone ($1.7\times10^{-2}$ M) was initially prepared. A 35 μL aliquot of this solution was placed in a 100 mL volumetric flask and the solvent was evaporated to air. The residue was then dissolved in either ultra-pure water (18.2 MΩ·cm—Millipore-filtered) or 0.9% w/v sodium chloride solution, resulting in a final concentration of pyrene of $6\times10^{-6}$ M. These solutions were then subsequently used as the solvent for the polymer solutions. Stock solutions of each Pluronic in water and saline solution were prepared. An aliquot of these solutions was dissolved in the pyrene/$H_2O$ or pyrene/saline solution. Solutions of different polymer concentration were obtained by diluting the stock polymer solution with the appropriate solvent. Mixed samples of two Pluronics were also prepared either with a fixed ratio of 1:1 or containing 0.01% L61. Samples were left to equilibrate for at least 3 hours prior to the experiment.

The fluorescence emission spectra were recorded on a Cary Eclipse fluorescence spectrophotometer (Varian, Oxford, UK) with $\lambda_{exc}$ = 340 nm. For the CMC, fluorescence intensities at 373, 384, 393 nm and, when it appeared, also at the excimer band centred at 490 nm, were measured. For each polymer, the critical aggregation concentration value was determined by using the intensity of the best resolved peak. At least two repeats were performed for each sample. Measurements were performed at 20˚C and 37˚C.

**Stability testing.**   The purpose of stability testing is to check whether pentamidine becomes altered with time under the influence of a variety of environmental factors such as temperature, humidity and light (Climatic zone IV, 30˚C and 65–75% relative humidity) [39].

In our initial 7 day assessment we also considered interaction of pentamidine with Pluronic as product-related factors may also influence its quality. A 5% or more change in initial content of pentamidine was considered significant. Pentamidine concentration at day 0, 10 and 7 was assessed by NMR.

A Bruker Advance 400 MHz spectrometer was used for recording the one-dimensional (1D) 1H NMR. Solutions of PTI, PTI/P85, PTI/P105 and PTI/F68 were prepared in $D_2O$ ($\geq$99.85% in deuterated component). Data were collected at days 0, 1 and 7. Samples were stored in amber NMR tubes at 37˚C.

**Partition coefficient determination.**   The partitioning coefficient, *P*, determines the fraction of drug incorporated into the micelle and provides thermodynamic characterization for the stability of the drug-micelle complex during dilution within the body fluids[11, 17].

The partition coefficient of pentamidine in the micellar core and bulk solvent, as described by Kabanov and co-workers [11], was measured for F68, P105 and mixtures of P105 and F68 (1:1), in both saline and aqueous solutions and at 20˚C and 37˚C.

Stock solution of $1\times10^{-6}$ M pentamidine isethionate salt (PTI) dissolved in water and in saline were prepared and were then subsequently used as the solvent for the polymer solutions and the preparation followed a similar method as for the CMC measurements. Samples were left to equilibrate for at least 3 hours prior to the experiment.

The fluorescence emission spectra were recorded on a Cary Eclipse fluorescence spectro-photometer (Varian, Oxford, UK) with $\lambda_{exc}$ = 260nm, for pentamidine. The fluorescence emission intensity at ca 340 nm was followed. The partition coefficient were calculated as described in Text B in S1 File.

**Drug release.** Solutions of Pluronic (1% F68 and 1% P105) with 10 mM PTI and PTI alone in water (2 mL) were loaded into 2 mL mini-dialysis tubes with 1 kDa molecular weight cut-off (GE Healthcare Bio-sciences Corp. USA). The tube was immersed in a 200 mL closed Duran flask which was placed in a water bath at 37°C for the duration of the experiment. Aliquots were collected from the immersion water (ultra-pure water (18.2 MΩ·cm—Millipore-filtered) in the flask every 30 min for the first 2 hours, every hour for the next 5 hours and then once more after 1 week. At the end of the experiment, an aliquot was collected from the dialysis cell. PTI concentrations were determined by UV spectroscopy (wavelength 260 mm).

The data was fitted to Ritger-Peppas model[40].

$$\frac{M}{M_\infty} = kt^n \qquad\qquad \text{Eq 1}$$

Where $M$ and $M_\infty$ are the cumulative amounts of drug released at time $t$ and at infinite time, respectively; k, the reaction constant, $t$ the time, $n$, the diffusional exponent describing the type of regime type: n = 1, case II transport, n = 0.5, Fickian diffusion, 0.5<n<1 non-Fickian diffusion.

**Dynamic light scattering (DLS).** Dynamic light-scattering (DLS) were performed with a photon correlation spectrometer Malvern Zetasizer Nano with a laser wavelength of 633nm. For obtaining the reduced scattered intensity, toluene was used as the standard and the increment in the refractive index, $\partial n/\partial c$, was assumed to be independent on the temperature and taken as 0.133 ± 0.002 mL·g$^{-1}$ [41]. The samples, of concentrations ranging between 1 to 5% w/v, were filtered prior to the measurements by 0.22 μm Millex syringe PVDF filters onto semi-micro glass cells. The temperature of the sample was controlled with 0.1°C accuracy by the built-in Peltier in the cell compartment. Size distributions were obtained for each sample from the analysis of the intensity autocorrelation function, which was performed with the Zetasizer software in the high-resolution mode to distinguish overlapping distributions.

**Small-Angle Neutron Scattering (SANS).** The architecture of the nanocarriers was measured by SANS on the LOQ instrument at ISIS pulsed neutron source (ISIS, Rutherford-Appleton Laboratory, STFC, Didcot, Oxford) (Text C in S1 File). The aggregation number ($N_{agg}$) and radius micellar size, including volume of core and shell region, correlates directly with are relevant to properties such as drug loading encapsulation efficiency, stability, half-life and hence circulation time[14].

**Simulations of Pluronic self-assembly and pentamidine encapsulation.** During this project, we worked to develop a model of the Pluronic and pentamidine systems that would allow us to simulate the self-assembly of the polymers and the encapsulation of the drugs. In order to simulate the timescales and system sizes required to study these systems, we utilized a coarse-grain approach; dissipative particle dynamics (DPD)[42]. This method has been used to study Pluronic before and has been shown to represent expected phenomena well. So we used the simulation parameters from [43].

## Evaluation of potential peripheral toxicity of pentamidine ± Pluronic

The toxicity of pentamidine in the presence of the Pluronic was explored using a variety of assays. The proposed route of administration for our Pluronic formulations with pentamidine was intravenous, hence the propensity for Pluronic to lyse red blood cells was studied using a

haemolytic assay (Text D in S1 File). Capillary wall integrity after exposure to the Pluronics was assessed using MDCK-MDR cells (see section 2.5b and Text E in S1 File). Peripheral toxicity of pentamidine/Pluronic formulations to the endocrine pancreas was evaluated by quantifying β-cell viability and insulin secretion from the mouse MIN6 β-cell line [44].

MIN6 β-cells were maintained in culture at 37˚C (95% air/5% $CO_2$) in DMEM supplemented with 10% foetal bovine serum, 2mM L-glutamine and 100U.ml$^{-1}$/0.1mg/ml$^{-1}$ penicillin / streptomycin, with a change of medium every 3 days. Cell were trypsinised (0.1% trypsin, 0.02% EDTA) when approximately 70% confluent and seeded into 96 well plates at a density of $3x10^4$ cells/well. After a 24 hour culture period to allow cells to adhere, the wells were washed with PBS and cells were pre-incubated for 2 hours in DMEM supplemented with 2mM glucose after which the medium was replaced with DMEM supplemented with Pluronic, pentamidine and Pluronic/pentamidine solutions in the presence of 2mM glucose. All tissue culture reagents were purchased from Sigma Aldrich (Poole, Dorset, UK).

The following formulations were evaluated: F68/PTI, P85/PTI and P105/PTI with Pluronic concentrations of 0, 0.01, 0.025, 0.1 and 0.5% w/v and PTI concentrations of 0, 1, 10 and 100 μM (20 formulations in total, including controls, Pluronic only, PTI only and solvent only were used). The cells were incubated under each treatment condition for 24 hours and then evaluated for their capacity to secrete insulin over an acute 30 minute incubation after which secreted insulin was quantified by RIA [45]. The effect of the formulations on β-cell viability was assessed by determining the access of trypan blue to the cell interior, indicative of a compromised plasma membrane[46].

## Blood-brain barrier studies

**Radiochemicals.** [$^3$H(G)]pentamidine (specific activity, 31.9 Ci/mmol; concentration, 10.74 μg/ml; radiochemical purity, 99.4%; MW 342.64) was custom synthesized and [$^{14}$C(U)] sucrose (specific activity, 536 mCi/mmol; concentration, 67.07 μg/ml; radiochemical purity, 98.7%) was purchased from Moravek Biochemicals, California, USA.

*In vitro* **permeability assays.** Several *in vitro* permeability models in both accumulation (reflecting plasma into the endothelial cell) and permeability (reflecting plasma to brain interstitial fluid) formats were evaluated for this study. This included Caco2 (permeability format), hCMEC-D3 (accumulation format), bEnd-3 (accumulation format) and MDCK-MDR (accumulation format) cell lines, before selecting the MDR1-MDCK cells (permeability format) as the most appropriate tool to address our objectives. MDR1-MDCK cells originate from transfection of Madin-Darby canine kidney (MDCK) cells with the MDR1 gene, the gene encoding for the human efflux protein, P-glycoprotein (P-gp). Using MDR1-MDCK cells avoids the complexities of multiple transporters by focusing specifically on P-gp.

*Preparation of formulation.* 1% (w/v) stock solutions of each Pluronic and 10 mM pentamidine isethionate were prepared in Hank's Balanced Salt Solution (HBSS) containing 25 mM HEPES and 4.45 mM glucose, at pH 7.4. These were further diluted to give final concentrations of 0.01, 0.1 or 0.5% (w/v) Pluronic containing 10 μM pentamidine isethionate. Formulations were stored at room temperature for 2–4 days prior to use.

*In vitro permeability assays.* MDR1-MDCK cells (NIH, Rockville, MD, USA) were maintained and permeability assays were performed at both Cyprotex (Macclesfield, Cheshire, UK) and King's College London. Analysis was by UPLC-MS/MS or liquid scintillation counting as appropriate.

Transmission electron microscopy confirmed appropriate cell morphology of a monolayer with microvilli on the apical membrane and Western blot confirmed expression of P-gp.

$3.4 x 10^5$ cells/cm$^2$ were seeded on Multiscreen plates with 0.4 μ polycarbonate Isopore membranes (Millipore, MA, USA) in DMEM/High glucose (Sigma-Aldrich, UK, D6429)

media containing 1% Non-Essential Amino Acids and 10% foetal calf serum (both from Sigma-Aldrich, UK). Plates were maintained at 37°C/5% $CO_2$ for 4 days before use. On the day of the assay, DMEM was removed and both the apical and basolateral surfaces of the cell monolayer were washed twice with transport medium consisting of HBSS containing 25 mM HEPES and 4.45 mM glucose, (pH 7.40; 37°C). Plates were incubated for 40 minutes at 37°C/5% $CO_2$ to stabilize physiological conditions. Transport buffer was removed from the apical or basolateral chamber and replaced with the formulation to be tested. Samples were taken from the apical and basolateral compartments after 1 hour of incubation at 37°C/5% $CO_2$. Samples, including the test formulation added to the apical chamber at t = 0 were analysed at Cyprotex using UPLC-MS-MS method (Text F in S1 File) to quantify the pentamidine isethionate content or were analysed for radioactivity using a Tricarb 2900TR liquid scintillation counter.

## *In situ* perfusions

The *in situ* brain/choroid plexus perfusion method for examination of the distribution of molecules into the brain and CSF is an established technique in the rat, guinea-pig and mouse [6, 47, 48]. It allows the passage of slowing moving molecules across the blood-brain and blood-CSF barriers to be examined and quantified in brain, capillary endothelial cells, and choroid plexus tissue for perfusion periods up to 30 minutes.

**Preparation of formulation.**    All formulations were prepared on the day of experiment at a Pluronic concentration of 0.1, 1.0 or 5% (w/v) using artificial plasma as a diluent. The artificial plasma consisted of a modified Krebs-Henseleit mammalian Ringer solution containing; 117 mM NaCl, 4.7 mM KCl, 2.5 mM $CaCl_2$, 1.2 mM $MgSO_4$, 24.8 mM $NaHCO_3$, 1.2 mM $KH_2PO_4$, 39 g dextran, 1 g/L of bovine serum albumin and 10mM glucose. [$^3$H(G)]pentamidine was added to give a final concentration of 157nM (equivalent to 5 μCi/ml). All formulations were stirred at room temperature for at least 1 hour to allow any chemical interactions and micelle formation to stabilize.

**Animal studies.**    All animal studies were performed within the framework of the Animals Scientific Procedures Act (1986) and Amendment Regulations 2012 and with consideration to the ARRIVE guidelines.

*BALB/c mice studies.* Adult male BALB/c mice were purchased from Harlan UK Ltd (Oxon, UK). All animals were maintained under standard temperature/lighting conditions and given food and water *ad libitum*. Only mice above 23g in weight were used for experiments. The study was approved by the King's College London Animal Welfare and Ethical Review Body.

*CD1 mice studies.* Adult female CD1 mice (20-25g) were purchased from Charles River (UK) for *in vivo* pharmacokinetic distribution studies. They were housed in specific pathogen-free individually vented cages and fed *ad libitum*. The experimental protocol was carried out with the approval of the London School of Hygiene & Tropical Medicine Ethics Committee. The protocol was reviewed and approved by the LSHTM Animal Welfare and Experimental Research Board.

## *In situ* perfusions

[$^3$H(G)]pentamidine formulations were delivered to the brain using an *in situ* brain perfusion technique as previously described [6]. Briefly, mice were anaesthetized (mixture of 2 mg/Kg Domitor/150 mg/Kg ketamine administered via the intraperitoneal route) and heparinized (100 U ip.). Oxygenated artificial plasma (described above) at 37°C was pumped via a 25-gauge cannula into the left ventricle of the heart, with the right atrium severed to prevent recirculation. Pumps were calibrated to deliver an overall flow rate of 5 ml/min from the cannula. [$^3$H(G)]pentamidine formulations (maintained at room temperature) were fed into the

flow line from a dual syringe infusion pump (Harvard Apparatus, UK), at a rate of 0.5 ml/min such that the formulation was diluted 1/10 immediately prior to entering the heart. 11 µM [$^{14}$C (U)]sucrose in artificial plasma (equivalent to 5 µCi/ml) was simultaneously fed into the flow line from a second identical syringe using the same pump set at 0.5 ml/min (equivalent to 1.1 µM or 0.5 µCi/ml entering the heart from the cannula). The perfusion was terminated at 10 minutes or 30 minutes, and the brain was sectioned as previously described [6]. Samples taken were those known to be invaded by parasites during second stage sleeping sickness and/or those which control mechanisms that are disrupted by the disease such as the sleep/wake cycle [6]. After the required samples were taken, the remaining brain tissue was homogenized and analyzed by the capillary depletion method described by Thomas & Segal [47](Text G in S1 File). All samples were solubilized with 0.5 ml Solvable (PerkinElmer Life and Analytical Sciences, Buckinghamshire, UK) for 48 hours. Scintillation fluid (3.5 ml Luma Safe, PerkinElmer Life and Analytical Sciences) was added and radioactivity ($^3$H and $^{14}$C) was counted on a Packard Tri-Carb2900TR scintillation counter in dual-label mode.

## Expression of results

The radioactivity (either $^3$H or $^{14}$C) present in tissue samples (dpm/g) was expressed as a percentage of that measured in the artificial plasma (dpm/ml) and was termed $R_{TISSUE}$%, as previously described [6]. Where stated, measurements for [$^3$H(G)]pentamidine were corrected for the contribution of drug present in the vascular space by subtraction of the $R_{TISSUE}$% for [$^{14}$C (U)]sucrose from the $R_{TISSUE}$% of [$^3$H(G)]pentamidine and these corrected values were termed $R_{CORR\ TISSUE}$%.

## Pharmacokinetic brain distribution experiments

***In vivo* pharmacokinetic experiments with [$^3$H(G)]pentamidine.**    Formulations containing 0.025% F68 with 8 µM [$^3$H(G)]pentamidine, 0.5% F68 with 8 µM [$^3$H(G)]pentamidine and 8 µM [$^3$H(G)]pentamidine alone were prepared in 0.9% sterile saline and allowed to equilibrate at room temperature for at least 1 hour before use. A 200 µl bolus of the formulation to be tested (equivalent to 15 µCi [$^3$H(G)]pentamidine) was administered to mice via the tail vein. At 2 hours post-injection, mice were exsanguinated via the right atrium of the heart into a heparinised syringe then perfused for 2.5 minutes with [$^{14}$C(U)]sucrose (1.1 µM, 0.5 µCi/ml) via the left ventricle, (all mice were anaesthetised with Domitor/ketamine and heparinised 20 minutes prior to exsanguination). Whole blood samples were immediately centrifuged for 15 minutes at 5,400 × g to remove red blood cells and the resulting plasma was placed on ice. A CSF sample was taken from the cisterna magna, the IVth ventricle choroid plexus and pituitary gland were collected and the brain was sectioned into right brain and left brain (both comprising frontal cortex and caudate putamen), cerebellum and midbrain (including pons and hypothalamus). The remaining brain (including occipital cortex and hippocampus) was used for capillary depletion analysis and all brain, circumventricular organs (CVO) and plasma samples were solubilized and subjected to dual label ($^3$H/$^{14}$C) scintillation counting as previously described.

***In vivo* pharmacokinetic experiments with pentamidine isethionate.**    Adult female CD1 mice (20-25g) were injected intravenously with pentamidine isethionate (4 mg/kg in 0.9% physiological saline) in the absence and presence of concomitant dosing with F68 (initial plasma concentration, calculated by estimating plasma volume at 10% of body weight) at 0.025%. Each group had an n = 3. Blood (<10 µl) was collected using a heparinized syringe at 1, 30, 120, 600 minutes post-injection and plasma prepared. Both blood and plasma samples were snap frozen on dry ice and stored at -80˚C before analysis. After the last blood sample,

the mice were perfused with sterile 0.9% physiological saline (via the hepatic portal vein), the brains removed, weighed and snap frozen. Analysis of samples was by a validated weak cation exchange solid phase extraction (WCX-SPE) approach performed by a specialist contract research organization (Cyprotex). Briefly samples were diluted with water, WCX-SPE sorbent was primed with MeOH and then water (to ensure phase was fully ionised). Samples were then loaded onto sorbent and washed with pH7 buffer and MeOH. Pentamidine was then washed off sorbent by eluting with a combination of MeOH/$H_2O$ + 5% v/v formic acid. If necessary, samples were then evaporated to dryness and reconstituted in injection solvent. Samples were analysed by UPLC-MS/MS as described above. LLOQ in plasma samples was 2 ng/ml and in brain samples was 80 ng/ml.

Additional experiments revealed that intravenous administration of 10mg/kg pentamidine isethionate plus or minus 0.5% F68 was toxic to the mice and the experiment was terminated.

**Data analysis.** All data are presented as means ±S.E.M and statistical analysis was carried out using Sigma Stat software, version 12.0 (SPSS Science Software UK Ltd, Birmingham, UK).

## Trypanocidal activity *in vitro*

*In vitro* activity of drug formulations against *Trypanosoma brucei* blood stream form trypomastigotes was determined *in vitro* using Alamar Blue (resazurin: Bio-Source, Camarillo, CA) as described by [49]. Prior to determination of the trypanocidal activity of Pluronic-pentamidine combinations, the $IC_{50}$ values of the Pluronic alone was established. Each Pluronic was tested in a 3-fold serial dilution in triplicate and in three separate experiments (n = 3). The diluent was HMI-9 media (Invitrogen, UK). Blood stream form *T. b. brucei* (strain S427) trypomastigotes, cultured in modified HMI-9 media supplemented with 10% v/v heat-inactivated foetal calf serum, (hi-FCS, Gibco, Life Technologies, UK), were incubated (37°C; 5% $CO_2$) at a density of $2 \times 10^4$/ml in the presence of pentamidine alone or pentamidine-Pluronic formulations for 66h. Resazurin (20 μl 0.49mM in PBS) solution was then added to each well and incubation continued for 6 hours. After incubation, samples were removed and fluorescence was measured using excitation 530nm and emission 590nm on a Spectramax M3 plate reader (Molecular Devices, USA). $IC_{50}$ values were determined (where appropriate) using GraphPad Prism.

# Results

## Evaluation of potential neurotoxicity of pentamidine

**Literature review.** We conducted a brief review of the literature to assess the potential neurotoxicity of pentamidine. Information was considered relevant to the NanoHAT project if it described an activity that could be detected in a simple profiling screen, rather than secondary readouts (e.g. hERG-mediated, downstream effects on cardiomyocyte [$Ca^{2+}$]i).

Table 2 lists the known pharmacology and approximate affinities of the interaction that have been reported for this compound. As the trypanocidal activity of pentamidine occurs at around 10 nM *in vitro*[37], we considered that any affinity greater than 1 μM (i.e. more than 100-fold greater than the trypanocidal concentration) was unlikely to be relevant.

There are 3 major target families for which pentamidine has significant affinity (<20 fold above trypanocidal range) that were of concern: the imidazoline$_2$ receptor (responsible for effects on central blood pressure control and pancreatic beta cells); inward rectifying (IR) potassium channels particularly blockade of Kir2.1 (this is more likely cardiac than CNS-relevant) and NMDA glutamate receptors.

**A neurological profiling screen.** A wide ligand profiling screen was carried out against 40 CNS targets (Perkin Elmer customised CNS screening; listed in Table A in S1 File), testing

**Table 2. Reported Pharmacology of Pentamidine *in vitro*.**

| Property | Affinity (μM) | Comments | Reference |
|---|---|---|---|
| Trypanocidal | 0.01 | Time-dependent | [37] |
| Imidazoline$_2$ receptor | 0.014 | 3H-idazoxan binding | [50] |
| Potassium channel expression/function | 0.17 | K(v)11.1(hERG) expression, K(IR)2.1 block | [51, 52] |
| NMDA (Ionotropic) glutamate receptor | 0.2 | Voltage dependent | [53] |
| Human anti-platelet | 1.1 | Inhibits fibrinogen binding to GP11b/IIIa | [54] |
| Rat NMDA receptor | 1.8 | Rat brain membrane 3H-dizocilpine binding | [55] |
| PRL phosphatases | 3 | Oncology target | [56] |
| Delta2glutamate receptor | 5 | Voltage independent | [53] |
| Calmodulin antagonist | 30 | Inhibits nNO synthase *in vitro* | [57] |
| Acid sensing ion channels (ASIC) | 38 | Potency 1b>3>2a>or = 1a | [58] |
| Serine proteases | 4000 | | [59] |

at a single pentamidine isethionate concentration of 10 μM (1000-times the trypanocidal concentration), with follow up concentration-response curves in any assay where there was greater than 70% inhibition. Pentamidine was inactive at 29 out of 40 CNS targets (including 5 glutamate receptor binding sites) at 10 μM and was re-tested against the remaining targets at a range of concentrations to generate an inhibitory constant, $K_i$ and this value was compared to trypanocidal activity (Table 3).

Selectivity screening of pentamidine identified 5 targets (imidazoline $I_2$ receptor; monoamine oxidase A and B; adrenergic $\alpha_1$ receptor; muscarinic receptor) for which it has significant affinity, and which should be monitored as we progressed through the screening cascade. In particular, pentamidine's high affinity for the imidazoline receptor may explain the cardiovascular adverse events associated with this drug. The project team considered that remaining targets were of minor concern, as the adverse events of drugs targeting the adrenergic monoamine oxidase and muscarinic systems are reasonably well described. The relatively low affinity of pentamidine for the remaining targets (histamine $H_2$ receptor; opioid receptor; adrenergic $\alpha_2$, $\beta$ receptors; 5HT transporter) indicated that the drug was unlikely to have significant effects until plasma/brain levels exceeded ~ 100-fold the trypanocidal concentration.

**Table 3. $K_i$ Values for Pentamidine Determined for Selected CNS targets together with the relative selectivity value when the $K_i$ is compared to trypanocidal activity ($IC_{50}$).These results, together with the calculated relative selectivity values compared with trypanocidal affinity, are listed in Table 3.**

| Target | $K_i$ (μM) | Relative to trypanocidal activity |
|---|---|---|
| Trypanocidal Activity | 0.01 | 1.0 |
| Imidazoline $1_2$ | 0.001 | 0.1 |
| Monoamine oxidase B | 0.181 | 18 |
| Monoamine oxidase A | 0.217 | 22 |
| Adrenergic alpha1 | 0.273 | 27 |
| Muscarinic (central) | 0.281 | 28 |
| Histamine H2 | 7.21 | 721 |
| Opioid | 1.41 | 141 |
| DA transporter | 2.11 | 211 |
| Adrenergicalpha2 | 10 | 1000 Estimate from single-point screen |
| Adrenergic β | 10 | 1000 Estimate from single-point screen |
| 5HT transporter | 10 | 1000 Estimate from single-point screen |

**Ion channel screen.**   We carried out ion channel screening at Chantest to investigate the potential potassium (K(IR)2.1) blocking liability reported by de Boer et al., (2010) (Table 2). Pentamidine isethionate salt was evaluated at 0.001, 0.01, 0.1, 1 and 10 µM (Table B in S1 File). The $IC_{50}$ value for pentamidine isethionate salt could not be calculated as the highest tested concentration resulted in hKir2.1 inhibition less than 50% (i.e. 12.3±1.3%). The $IC_{50}$ is estimated to be greater than 10 µM. The positive control (100 µM barium) confirms the sensitivity of the test system to ion channel inhibition.

## Formulation development

As this was a milestone driven project an iterative, dynamic approach was utilized to select the lead formulation to take forward as quickly as possible in the screening cascade (Fig 1), hence not all Pluronic formulations were assessed with each of the methods.

**Phase Behaviour.**   L61 phase diagrams were evaluated by visual inspection from 20˚C to 50˚C for L61 alone and in mixtures with P105 and/or F68 in water and saline solutions. L61 presents a cloud point around 24˚C [60] and F68 does not improve its solubility, while P105 does to some extent (Tables C and D in S1 File).

**Critical micelle concentration (CMC) by fluorescence spectroscopy.**   CMC were measured for individual Pluronic and mixtures of F68, P85, P105 and L61 at 20˚C and 37˚C, both in aqueous and saline (0.9 wt%) solutions, using the intensity of pyrene fluorescence emissions (Table 4; Fig B in S1 File). Mixtures of two Pluronics in both aqueous (aq) and saline (sal) mediums were prepared in either a fixed mass ratio of 1:1 or with the addition of 0.01% w/v L61 and the CMC determined. All CMC curves show two inflection points, a feature widely reported in the literature; the first corresponds to the onset of aggregation and was chosen as the CMC (Fig B in S1 File; Table 4), giving the following values in saline solution at 37˚C (g/L):

**Table 4. CMC Values of Pluronic Dissolved in Pure Water (aq) or Saline (sal) at 20˚C and 37˚C Determined Using Pyrene Fluorescence Intensity.**  Values Mean ± S.D. Saline (0.9 wt%).

| Temperature Sample | 20˚C g/L | 37˚C g/L |
|---|---|---|
| | CMC | CMC |
| P85aq | 0.320±0.007 | 0.043±0.007 |
| P85sal | 0.146±0.031 | 0.042±0.018 |
| F68aq | 0.274±0.031 | 0.061±0.004 |
| F68sal | 0.273±0.003 | 0.048±0.012 |
| P105aq | 0.243±0.0140 | 0.073±0.014 |
| P105sal | 0.190±0.0093 | 0.069±0.019 |
| L61 aq | 0.030±0.032 | n.a. |
| L61 sal | 0.0240±0.024 | n.a. |
| **Fixed ratio 1:1 mixture** | | |
| P85+F68aq | 0.742±0.000 | 0.095±0.000 |
| P85+F68sal | 0.678±0.000 | 0.099±0.000 |
| P85+L61 aq | 0.268±0.000 | n.a. |
| P85+L61 sal | 0.3024±0.000 | n.a. |
| **Sample + L61 (0.01w/v%)** | | |
| P85aq | 0.114±0.004 | 0.051±0.0264 |
| P85sal | 0.284±0.128 | 0.0734±0.032 |
| F68aq | 0.201±0.004 | 0.051±0.018 |
| F68sal | 0.206±0.028 | 0.043±0.000 |
| P105 aq | 0.242±0.030 | 0.070±0.024 |
| P105 sal | 0.194±0.014 | 0.0833±0.048 |

$P85_{sal} = 0.042 \pm 0.018$; $F68_{sal} = 0.048 \pm 0.012$ and $P105_{sal} = 0.069 \pm 0.020$. Overall, these CMC values are fairly similar and do not allow a prioritisation based on CMC alone. The CMC of F68 and P85 mixtures (1:1 mass ratio) is about double the CMC, when expressed in total Pluronic mass, of the individual polymers suggesting the absence of mixed micelles in these mixtures. Small amounts of L61 (0.01%w/v) does not affect the CMC of F68 or P85 or P105 under the conditions tested.

**Stability of the formulations.**    Pentamidine stability in solution was followed by NMR. Pentamidine and pentamidine/Pluronic solutions prepared in $D_2O$ were kept in amber NMR tubes at 37˚C. Spectra were measured at days 0, 1 and 7. As a control, pentamidine in $D_2O$ was left at 4˚C and measured at day 0 and 7. NMR data showed no significant change on peak position or peak intensity when compared to day 0 measurements or to control samples, confirming no thermal degradation of pentamidine after 7 days at 37˚C.

**Partition.**    Partition of PTI in the micelles was measured by fluorescence spectroscopy for P105 and F68. Pentamidine has a slightly larger partition coefficient in F68 than in P105 (Table 5). Measurements in mixtures (F68/L61, P105/L61 and F68/P105, 1:1 mass ratio in all cases) do not significantly change the partition coefficient.

The values of Log P obtained in saline and aqueous solutions are rather similar, suggesting that pentamidine partition is not sensitive to the saline levels used here.

The effect of temperature is quite weak (Table 5), and does not follow the same trend with the two Pluronic studied: values of LogP for P105 are lower at 20˚C than at 37˚C (but still very close); instead, for F68 the partition of PTI decreases slightly at higher temperature.

At biologically relevant concentrations, 0.5 wt% Pluronics and $1.0 \times 10^{-6}$ M PTI, extrapolation of the Log P data suggests that ca. 0.1 PTI molecules would be incorporated in one P105 micelle, and 0.01 PTI molecules in one F68 micelle. At the concentrations used for SANS (5 wt% Pluronic and 1 wt% PTI), extrapolating these numbers give 166 PTI molecules in the micellar core per P105 micelle and 15 for F68 micelle.

The relative low values of log P for PTI/Pluronic system (for comparison log P for pyrene/Pluronics is ca. 2.5 and 3.5 for F68 and P105, respectively [11]), is not surprising given the high water solubility of pentamidine, and helps to explain the drug release profile for PTI / Pluronics systems discussed next.

Overall, this means that Pluronic have a limited capacity to interact with pentamidine and prolong its circulation.

**Drug release.**    Solutions of 10 mM pentamidine or 10 mM pentamidine plus 1% F68 or 1% P105 were loaded in dialysis cells and the amount of pentamidine eluting from the cells into water at 37˚C were measured over time (Fig D in S1 File). Both reaction type and reaction

**Table 5. The fraction of pentamidine incorporated into the Pluronic micelle expressed as a partitioning coefficient, P.** The Pluronic was dissolved in pure water (aqueous) or saline (saline) at 20˚C and 37˚C. (Also see Fig C in S1 File).

| Pluronic | Solvent | Temperature ˚C | Log P |
|---|---|---|---|
| P105 | saline | 20 | 1.06 |
| | | 37 | 1.15 |
| P105 | water | 20 | 0.99 |
| | | 37 | 1.09 |
| F68 | saline | 20 | 1.67 |
| | | 37 | 1.47 |
| F68 | water | 20 | 1.67 |
| | | 37 | 1.46 |

                    

**Table 6. Geometric parameters from model-fitting of the SANS Pluronic data at 37˚C, including core and shell micellar sizes, fraction of solvent in the corona ($\chi_{solv}$) and aggregation number ($N_{agg}$).** (Also see Fig E in S1 File).

| Sample | Core radius (Å) | Shell thickness (Å) | Total radius (Å) | $\chi_{solv}$ | $N_{agg}$ |
|---|---|---|---|---|---|
| F68 5% | 15.4 | 36.5 | 52.0 | 0.99 | 2.37 |
| F68 5%/ PTI 1% | 15.1 | 34.7 | 49.8 | 0.98 | 2.25 |
| F68 5%/ PTI 3% | 15.5 | 33.9 | 49.4 | 0.99 | 2.38 |
| P85 5% | 42.9 | 31.4 | 74.3 | 0.95 | 35.4 |
| P85 5%/ PTI 1% | 41.5 | 30.5 | 72.0 | 0.95 | 32.4 |
| P85 5%/ PTI 3% | 41.0 | 30.5 | 71.5 | 0.99 | 31.6 |

constant, for PTI alone and PTI/Pluronic were in a similar range. ca. 0.5 (Fickian diffusion) for reaction type and ca 0.3 for reaction constant. No significant difference was observed between PTI/Pluronics and PTI/water systems. Thus, in the conditions tested, pentamidine release seems to be dominated by diffusion and Pluronic micelles were not a barrier for drug release.

**Aggregation number and Micellar size:.** Pluronic micelles can be reasonably described as a compact core formed by a dry PPO block surrounded by a highly hydrated shell formed by the two PEO blocks[61, 62]. The core-shell model was thus used to provide a more detailed characterisation of the morphology of the Pluronic micelles in $D_2O$ and how it is affected by the presence of PTI, using input values for the core radius and shell thickness were based on hydrodynamic radius values obtained by DLS (Table E in S1 File). A term to compensate for polydispersity was included for both Pluronics, as well as a structure factor ($S(q)$), corresponding to a hard sphere model, in order to account for intermicellar interactions. A summary of the main parameters obtained from the analysis of the data (Fig E in S1 File) is present in Table 6.

A direct comparison of F68 and P85 micelles in $D_2O$ shows that both have similar shell thickness, with F68 showing values slightly larger, 36.5 Å vs 31.4 Å, respectively. It is worth noting that F68 EO blocks have on average 76.4 EO units while P85 blocks are only 26.1 units long. The core of F68 micelles are significantly smaller than P85 micelles, 15.4 vs 42.9 Å. In terms of PO content, the F68 PO block is 29 units long while P85 is 40 units long. Overall, P85 micelles are larger than F68 micelles, 74.3 vs 52.0 Å, respectively.

The molecular dynamic simulation work agrees well with these experimental results. The average aggregation number per micelle ($N_{agg}$) and the average number of micelles ($N_{mic}$) were calculated once the systems had equilibrated have been measured. Fig 2 shows plots of $N_{agg}$ and $N_{mic}$ as a function of Pluronic concentration for both the F68 and P105 Pluronics. We carried out simulations over a range of Pluronic concentrations that span the CAC and the CMC values observed experimentally to validate the models (at least qualitatively). From Fig 2, one can see that in both systems, once we have passed the CAC the number of micelles remains more or less constant but they continue to grow in size as the concentration increases until we reach the CMC at which point the size of the micelles more or less plateaus. Also, when comparing the behavior of the F68 and P105 Pluronics, we found that the P105 Pluronics form larger aggregates when near the CMC as compared to that for the F68 Pluronics, and therefore fewer micelles. Note, we have also used molecular dynamic simulations to examine mixtures of F68 and L61 Pluronics, and the results of those systems are presented in Fig F in S1 File.

Additionally, we have compared the findings from the simulations with the identified values (dashed lines) of the CAC and CMC from the experimental systems.

In the presence of 1% PTI, a small reduction in size was observed for both Pluronics, ca. 2 Å in both cases. The increase to 3% PTI does not cause further changes.

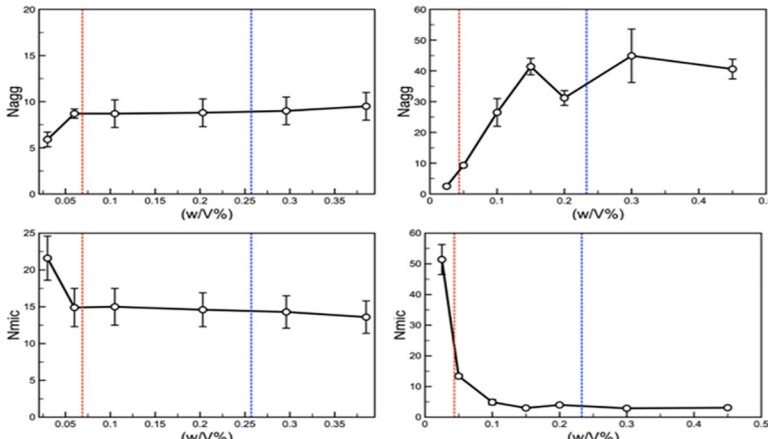

**Fig 2. The average number of Pluronic molecules found in a micelle ($N_{agg}$) and the number of micelles in our system (after they have equilibrated) ($N_{mic}$) as a function of the concentration of the Pluronics in the system for both the F68 (left) and P105 (right) Pluronics.**

The coronas were highly hydrated, as reported for these polymers [63, 64]. F68 micelles were more hydrated than P85: for each EO unit in the shell, there were 17 $D_2O$ molecules in a F68 micelle but only 3.4 in a P85 micelle.

The addition of pentamidine leads to a subtle, but perceptible, reduction of the number of water molecules in the F68 micelle shell. For P85, no measurable changes were observed.

## Peripheral toxicity

Pluronic concentrations used in the biological assays were based on the CMC measurements. Peripheral toxicity of the individual polymers was assessed. L61 was not studied at this stage due to its limited solubility. The results of the haemolysis and capillary integrity studies are found in Text H and I in S1 File.

**Effect of Pluronics on insulin secretion and beta-cell viability.** Exposure of MIN6 β-cells to 1, 10 and 100μM pentamidine for 24 hours caused a concentration-dependent inhibition of acute insulin secretion (Fig 3). Surprisingly, P85 and 105 were significantly more

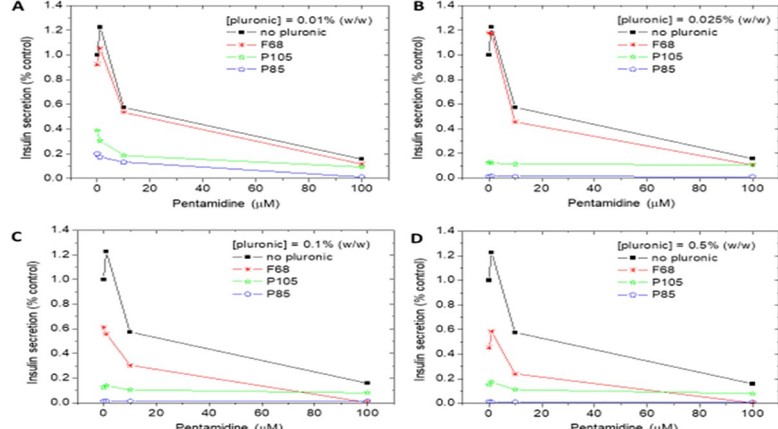

**Fig 3. The effect of pentamidine and Pluronics on insulin secretion from MIN6 β-cells.** (A-D) P85 and P105 induced a strong suppression of insulin secretion from MIN6 β-cells even at low concentrations. (C-D) F68 only induced insulin secretion suppression at concentrations ≥0.1% w/v. Data are expressed as a percentage of insulin secretion from MIN6 β-cells incubated in the absence of pentamidine or Pluronics.

**Table 7. The Inhibitory Concentration (IC$_{50}$) required to reduce number of bsf trypomastigotes by 50%.** Pluronic were tested at 12 serial dilutions in triplicate and repeated in 3 separate experiments (n = 3) to produce IC$_{50}$ values.

| w/v % | F68 | F68/0.01% L61 | P85 | P105 |
|---|---|---|---|---|
| IC$_{50}$ | 0.48% | 0.46% | 0.00021% | 0.00084% |
| 95% CI | 0.38–1.35 | 0.027–0.94 | 0.00056–0.0014 | 0.00070–0.0012 |

effective than pentamidine in inhibiting insulin secretion, such that insulin release was substantially inhibited by these Pluronics in the absence of pentamidine at all concentrations tested (0.01–0.5% w/v) (Fig 3A-D). Low concentrations of F68 (0.01 and 0.025% w/v) generated similar inhibitory effects on insulin secretion as unformulated pentamidine (Fig 3A and 3B) and increased toxicity was observed with higher concentrations of F68 (Fig 3C and 3D).

Trypan blue staining indicated that the MIN6 β-cells were able to tolerate pentamidine concentrations of 1 and 10 μM, but 100 μM pentamidine, which induced maximal inhibition of insulin secretion, was accompanied by a large number of cells taking up Trypan blue (Figs G and H in S1 File). These micrographs are indicative of the suppression of insulin secretion by pentamidine being associated with marked reductions in β-cell viability, but the plasma membrane was largely intact as there was no leakage of insulin, a 5.5 kDa peptide, from the cell interior. The combination of 100 μM pentamidine with 0.5% w/v F68, which caused maximal suppression of insulin release (Fig 3), led to the highest proportion of cells that showed Trypan blue staining.

## Trypanocidal activity in vitro

*The In vitro* activity of Pluronic drug formulations alone against *T. b. brucei* blood stream form trypomastigotes was determined showing low trypanocidal activity of F68 compared to high activity of P85 and P105 (Table 7).

In further studies the anti-trypanosomal activity of combinations of F68 and pentamidine were assessed (Table 8)(74). The IC$_{50}$ (± 95% CI) values of pentamidine were 2.11 x 10$^{-5}$ ± (1.79 x 10$^{-5}$–2.50 x 10$^{-5}$) μM alone, 6.36 x 10$^{-6}$ (± 4.43 x 10$^{-6}$–9.12 x 10$^{-6}$) μM with 0.01% F68 and 3.25 x 10$^{-6}$ ± (3.13 x 10$^{-7}$–3.38 x 10$^{-5}$) μM with 0.001% F68.

To determine if the addition of Pluronic to pentamidine had an additive effect on the trypanocidal activity of pentamidine, it was decided that work should focus on F68 rather than the other Pluronics, as both P85 and P105 caused an inhibitory effect on insulin secretion. Although IC$_{50}$ values could only be determined for two combinations, in part due to the high starting concentration of pentamidine used, a limited interaction between Pluronic F68 and

**Table 8. The % of bsf trypomastigotes inhibited by pentamidine/pluronic combinations.** The combination formulation was tested in triplicate and repeated in 3 separate experiments (n = 3).

| | Pentamidine (μM) | | | | | |
|---|---|---|---|---|---|---|
| | 1 | 0.3 | 0.000152 | 5.1 x 10$^{-5}$ | 1.7 x 10$^{-5}$ | 5.7 x 10$^{-6}$ |
| F68 (w/v %) | | | | | | |
| 0.5% | 99.5% | 98.6% | 98.6% | 98.3% | 98.3% | 99.2% |
| 0.1% | 98.5% | 97.7% | 97.1% | 97.1% | 97.3% | 97.7% |
| 0.025% | 98.3% | 97.5% | 97.0% | 96.9% | 97.0% | 90.6% |
| 0.01% | 98.4% | 97.6% | 96.4% | 95.1% | 82.8% | 3.4% |
| 0.001% | 98.3% | 97.4% | 96.4% | 91.9% | 73.1% | 1.8% |
| 0% | 98.3% | 97.4% | 92.7% | 65.3% | 35.0% | 4.1% |

**Table 9. The Effect of P85, F68 and P105 on the Apparent Permeability of [$^3$H(G)]pentamidine (9 nM) MDR1-MDCK Cell Monolayers in the Apical to Basolateral Direction and the Basolateral to Apical Direction.** The percentage recovery of pentamidine is also shown. All the data has been corrected for extracellular space by subtracting [$^{14}$C(U)]sucrose (5.5 μM) $P_{app}$ values which ranged from 0.89 to 2.00 x $10^{-6}$ cm/s. Each value represents three replicates for each n and n = 3. n.d. = not determined as integrity of the barrier compromised.

| [$^3$H(G)]Pentamidine (9 nM) | Pluronic Concentration (%) | $P_{app}$ A2B ($10^{-6}$ cm/s) | $P_{app}$ B2A ($10^{-6}$ cm/s) | A2B (%) | B2A (%) |
|---|---|---|---|---|---|
| | | Mean±SEM | Mean±SEM | Mass balance | Mass balance |
| | 0 | 0.678±0.025 | 0.776±0.062 | 84 | 85 |
| | 0.01% P85 | 0.310±0.142 | 0.431±0.161 | 86 | 87 |
| | 0.1% P85 | 0.561±0.0.172 | 0.227±0.081 | 89 | 89 |
| | 0.5% P85 | n.d. | n.d. | 90 | 90 |
| | 0.01% P105 | 0.577±0.0710 | 0.818±0.086 | 86 | 89 |
| | 0.1% P105 | 0.898±0.161 | 0.776±0.054 | 89 | 88 |
| | 0.5% P105 | n.d. | n.d. | 91 | 91 |
| | 0.01% F68 | 0.200±0.115 | 0.106±0.061 | 95 | 83 |
| | 0.1% F68 | 0.221±0.067 | 0.033±0.019 | 98 | 87 |
| | 0.5% F68 | 0 | 0 | 98 | 84 |

pentamidine was observed at the lowest F68 concentrations (Table 8 boxes shaded in red), suggesting that the addition of Pluronic had an additive effect on the trypanocidal activity.

## Blood-brain barrier: *In vitro* permeability assays

We examined the ability of different pentamidine-Pluronic formulations to cross the BBB using the MDR1-MDCK cell line. Two analytical methods were applied: one detected pentamidine isethionate using UPLC-MS/MS (Table F in S1 File) and the other detected radiolabelled pentamidine using liquid scintillation counting (Text J in S1 File and Table 9). The presence of the Pluronics (F68, P105 or P85) at concentrations of 0.01% and 0.1% did not significantly increase the distribution of pentamidine isethionate or [$^3$H(G)]pentamidine across the MDR1-MDCK monolayer measured over 60 minutes.

In conclusion, our target formulation characteristics of at least a 2-fold increase in pentamidine / pentamidine isethionate movement across the monolayer, compared with unformulated pentamidine, was not observed using these *in vitro* models of BBB permeability.

## Blood-brain barrier *In situ* brain perfusion

**Pluronic P85 and Pluronic P105.** Co-formulation of 15.7 nM [$^3$H(G)]pentamidine with Pluronic P85 did not significantly increase [$^3$H(G)]pentamidine accumulation in any of the brain regions examined using *in situ* brain perfusion (Table G in S1 File). Additional information regarding this data set can be found in the supplementary Text K in S1 File.

An overall decrease in the [$^{14}$C(U)]sucrose-corrected uptake of [$^3$H(G)]pentamidine into brain parenchyma was observed when 15.7nM [$^3$H(G)]pentamidine was co-formulated with 0.1% (p<0.001) and 0.5% (p<0.001) P105, as shown in Table H in S1 File, but (like P85) these data did not reach statistical significance in any of the individual regions sampled (Two-Way ANOVA with Bonferroni's pairwise comparisons).

In contrast, there was a 33% increase in the [$^{14}$C(U)]sucrose-corrected uptake of [$^3$H(G)]pentamidine into the endothelial cell pellet when it was co-formulated with 0.1% P105 (p = 0.027; Two- way ANOVA with Bonferroni's pairwise comparisons). This increase was apparent in only 3 out of 6 mice, and was associated with penetration of the brain tissue by the vascular space marker [$^{14}$C(U)]sucrose, perhaps indicating an increase in the permeability of the apical/luminal endothelial cell membrane. Additional information regarding this data set can be found in the supplementary Text L in S1 File.

## PLURONIC F68

*10 minute perfusions.* Co-formulation of [³H(G)]pentamidine with F68 resulted in an overall decrease in accumulation of [³H(G)]pentamidine into brain parenchyma after 10 minutes of perfusion (p = 0.002 for 0.1% and p = 0.03 for 0.5% respectively; Two-way ANOVA with Bonferroni's pairwise comparisons) (Table I in S1 File). A decrease in vascular space as measured by accumulation of [¹⁴C(U)]sucrose was also measured when 0.01 or 0.1% F68 (but not 0.5%) was present in the artificial plasma (p = 0.042 for 0.01% and p = 0.004 for 0.1% respectively; Two-way ANOVA with Bonferroni's pairwise comparisons) (Table J in S1 File).

F68 did appear to increase accumulation of [³H(G)]pentamidine into the endothelial cell pellet at concentrations of 0.01% and 0.1%, but these results did not attain significance. This increase in [³H(G)]pentamidine, did not appear to be associated with a concomitant increase in uptake of [¹⁴C(U)]sucrose (p>0.05) and might have been due, at least in part, to a small decrease in the amount of drug crossing the basolateral membrane to enter the brain parenchyma, as indicated by a marginal reduction of [³H(G)]pentamidine in the supernatant (Table I in S1 File).

Co-formulation of [³H(G)]pentamidine with 0.5% F68 resulted in a 2-fold increase in uptake into the pituitary gland after 10 minutes of perfusion (p = 0.017; 1-way ANOVA with Bonferroni's pairwise comparisons). A similar, but not statistically significant increase was observed in uptake of [¹⁴C(U)]sucrose into this organ over the same time period.

*30 minute perfusion.* Accumulation of [¹⁴C(U)]sucrose measured in brain parenchyma, as a percentage of concentration in the artificial plasma ($R_{TISSUE/PLASMA}$%), ranged from 1.3% in the hippocampus to 4.3% in the pons after 30 minutes of perfusion. These values are almost identical to our previously published data for BALB/c male mice (1.6 and 4.5% respectively)[7]. Accumulation of [³H(G)]pentamidine, when corrected for vascular space ranged from 6.9% in the hippocampus to 15% and 10.9% in the more highly vascularized regions of the hypothalamus and pons, respectively. These values were slightly higher than our previously published data (4.3% for hippocampus, 7.6% for hypothalamus and 8.2% for pons) and might reflect changes in expression of transporters due to differences in environment/diet or selective pressures during breeding.

Formulation of 15 nM [³H(G)]pentamidine with 0.01% or 0.1% F68 did not affect [¹⁴C(U)] sucrose brain space (p = 0.139 and 0.460 respectively; 2-way ANOVA with Bonferroni's post-tests). No significant differences were observed in [³H(G)]pentamidine accumulation at these concentrations (p = 0.120 and 1.000 respectively; 2-way ANOVA with Bonferroni's post-tests). Similarly, F68 had no significant effect on [¹⁴C(U)]sucrose or [³H(G)]pentamidine accumulation in the capillary depletion samples after 30 minutes of perfusion (p>0.05 for each concentration tested for each isotope; 2-Way ANOVA) nor in the circumventricular organs (p>0.05 for each concentration tested for each isotope; 2-Way ANOVA).

There was an approximate 2-fold increase in accumulation of both [³H(G)]pentamidine and the vascular space marker [¹⁴C(U)]sucrose in the brain parenchyma of mice that were perfused with formulations containing 0.5% F68, (p = 0.003 and p <0. 001 respectively; 2-way ANOVA with Bonferroni's post-tests), as shown in Tables K and L in S1 File. Visible signs of damage to the BBB including permeation and staining with Evans blue (MW 961), were also observed in some mice. The results from the capillary depletion analysis after 30 minutes of perfusion would also appear to reflect damage to both the apical and basolateral endothelial cell membranes, with a tendency for increased permeation of [¹⁴C(U)]sucrose into the brain parenchyma, as demonstrated by a small, though not statistically significant rise in this isotope being detected in the supernatant (Tables K and L in S1 File).

Co-formulation of [³H(G)]pentamidine and [¹⁴C(U)]sucrose with 0.5% F68 resulted in an increase into the pituitary gland and the choroid plexus when the perfusion time was extended to 30 minutes, although these results were not statistically significant.

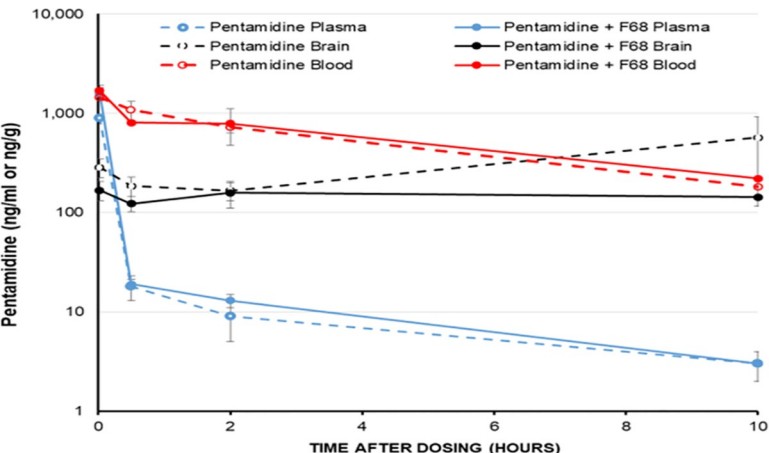

**Fig 4. The effect of Pluronic F68 on pentamidine concentrations in CD1 mouse plasma, blood and brain after an intravenous dose.** Each point represents an n of 3. 4mg/kg pentamidine ± 0.025% F68 i.v. Values ± SD.

### *In vivo* pharmacokinetic experiments with pentamidine isethionate or [³H (G)]pentamidine

F68 at the 0.025% does not change the accumulation of pentamidine isethionate in the plasma, brain parenchyma or blood in the CD1 mouse up to 10 hours post-dosing (Fig 4). There might be a late-onset increase in brain concentrations in the pentamidine alone group, but as the standard deviations for this group at this time-point are large this is unlikely to be significant.

Table 10 shows the mean plasma and CSF (corrected for blood/sucrose contamination) concentrations for [³H(G)]pentamidine and/or its metabolites, measured at 2 hours after intra-venous injection. No significant differences were observed when [³H(G)]pentamidine was co-formulated with either 0.025% or 0.5% F68 (p >0.05 for plasma and CSF; One-way ANOVA). Similarly, no significant differences were observed in uptake of [³H(G)]pentamidine or the vascular space marker [¹⁴C(U)]sucrose, into the brain parenchyma, capillary depletion samples or the circumventricular organs when [³H(G)]pentamidine was injected in the presence or absence of F68 (p>0.05; 2-way ANOVA with Bonferroni's pairwise comparisons) as shown in Table 10.

## Discussion

In this study we generated pentamidine/Pluronic formulations and prioritised 18 formulations using a rational, iterative approach (Fig 1). The milestones were intended to ensure that the most appropriate formulations, on the basis of *in silico* and *in vitro* data, were taken forward to the *in vivo* pharmacokinetic studies and that the formulations with the greatest likelihood of success would be assessed for toxicity issues *in vivo* and tested in animal efficacy models of stage 1 and stage 2 HAT. An ideal formulation for injection should be equipped with characteristics that improved the stability and safety profile of pentamidine, enhanced therapeutic effect, and accelerated the absorbance of drugs.

Since increasing the concentration of pentamidine in the brain may cause an intractable neurotoxicity and serious adverse events our empirical starting point was a customised, wide ligand profiling screen carried out against 40 CNS targets (Table 2 and Table A in S1 File). Five targets (imidazoline $I_2$ receptor; monoamine oxidase A and B; adrenergic $\alpha_1$ receptor; muscarinic receptor) were identified to have significant affinity for pentamidine (Table 3). All but one of these (imidazoline $I_2$ receptor) had a 20–1000 fold lower affinity than the relative

**Table 10. Uptake of [$^3$H(G)]pentamidine into brain tissue (corrected for vascular/[$^{14}$C]sucrose space) and CSF (corrected for blood/[$^{14}$C]sucrose contamination)at 2 hours post-injection in BALB/c mice.** Data is presented (a) as the tissue/plasma ratio and converted into concentrations in ng/g of tissue (b) and as concentration for the terminal plasma and CSF samples (c). A limitation of measuring pentamidine by scintillation counting is that any metabolites produced during the 2 hours that have retained the radiolabel, will be counted as [$^3$H(G)]pentamidine. These metabolites may have different transport characteristics and may or may not be active against trypanosomes.

| (a) | $R_{TISSUE/PLASMA}$% (mean±SEM) | | |
|---|---|---|---|
| Region | Control (15.7 nM pentamidine) (n = 6) | 0.025% F68 + (15.7 nM pentamidine) (n = 6) | 0.5% F68 + (15.7 nM pentamidine) (n = 5) |
| Right brain | 115.52 (± 12.46) | 120.29 (± 17.14) | 87.36 (± 20.36) |
| Left brain | 152.29 (± 33.48) | 111.85 (± 19.15) | 106.10 (± 12.92) |
| Cerebellum | 204.02 (± 35.28) | 208.87 (± 28.81) | 172.48 (± 30.34) |
| Midbrain | 181.18 (± 45.30) | 254.02 (± 35.48) | 180.00 (± 32.83) |
| Homogenate | 249.41 (± 35.59) | 184.18 (± 35.22) | 293.81 (± 122.95) |
| Supernatant | 123.35 (± 28.45) | 99.72 (± 9.02) | 98.66 (± 9.47) |
| Pellet | 479.72 (± 72.50) | 310.63 (± 38.62) | 536.52 (± 212.72) |
| Choroid plexus | 24666.66 (± 4928) | 19628.89 (± 4672) | 20463.70 (± 1827) |
| Pituitary gland | 15053.41 (± 3598) | 11285.42 (± 2008) | 15061.87 (± 5321) |
| (b) | Mean concentration (ng/g or ng/ml for the supernatant ±SEM) | | |
| Region | Control (15.7 nM pentamidine) (n = 6) | 0.025% F68 (15.7 nM pentamidine) (n = 6) | 0.5% F68 + (15.7 nM pentamidine) (n = 5) |
| Right brain | 0.363 (± 0.035) | 0.417 (± 0.061) | 0.302 (± 0.058) |
| Left brain | 0.472 (± 0.084) | 0.383 (± 0.063) | 0.375 (± 0.048) |
| Cerebellum | 0.607 (± 0.032) | 0.719 (± 0.097) | 0.591 (± 0.084) |
| Midbrain | 0.494 (± 0.075) | 0.866 (± 0.115) | 0.614 (± 0.072) |
| Homogenate | 0.820 (± 0.183) | 0.643 (± 0.132) | 0.988 (± 0.375) |
| Supernatant | 0.363 (± 0.037) | 0.345 (± 0.035) | 0.351 (± 0.043) |
| Pellet | 1.482 (± 0.151) | 1.067 (± 0.125) | 1.827 (± 0.662) |
| Choroid plexus | 74.68 (± 11.48) | 84.04 (± 5.78) | 72.20 (± 7.60) |
| Pituitary gland | 43.76 (± 3.82) | 37.58 (± 6.54) | 68.13 (± 15.05) |
| | Mean concentration | | |
| (c) | Control | 0.025% F68 | 0.5% F68 |
| CSF pg/ml (± SEM) | 2.669 (± 0.765) | 1.948 (± 0.826) | 3.592 (± 1.932) |
| Plasma ng/ml (± SEM) | 0.343 (± 0.061) | 0.345 (± 0.013) | 0.356 (± 0.026) |

trypanocidal activity and did not generate major concern[55]. The activity against the imidazoline I$_2$ receptor may explain the cardiovascular adverse events with this drug. We were unable to reproduce the result of De Boer et al., 2010[51] in a recombinant human system indicating that pentamidine was without effect (at up to 10 μM) on the hKir2.1 potassium channel-induced inward rectifying current (Table 2 and Table B in S1 File). Thus progression could continue through the screening cascade.

For the Pluronics tested in this study (P85, P105, F68 and L61), phase behaviour [38, 65] and cloud points [66] are well established. P85, P105 and F68 are soluble in water and saline solutions at both 24°C and 37°C. L61 has a very low cloud point at 24°C. Pure L61 therefore has limitations as a formulation for drug delivery. Our phase diagrams revealed that F68, which is highly hydrated, is unable to improve the solubility of highly hydrophobic L61 to a great extent, so it was not possible to pursue a 1:1 mixture of L61:F68 in the assays (Tables C and D in S1 File).

Using molecular dynamics simulations and physical techniques, we elucidated the structural properties of Pluronic P85, P105, F68 and L61 micelles, and were able to extract fundamental parameters required for biological evaluation of the formulations. For example, the CMC were measured for F68, P85 and P105 at 20°C and 37°C both in aqueous as well as saline (0.9 wt%) solutions. Several values for the CMC of Pluronics can be found in the literature [11,

67–70]. These values tend to vary widely, showing as much as one order of magnitude differences for the same Pluronic[71]. This has been attributed to several reasons: difference in molecular weight distribution between batches [70, 72], presence of impurities such as diblocks[72, 73] and differences inherent to the technique employed[74]. In addition, for some Pluronic systems, two critical concentrations are detected, both in surface tension and spectroscopic experiments [68, 72]. This behaviour has been ascribed to formation of premicellar aggregates occurring before full micelle formation[67, 68, 75–77]. In this work, which used the intensity of pyrene fluorescence emission, two critical concentrations were also detected (Fig B in S1 File). The CMC values presented here (Table 4) are taken from the first break point. The CMC values achieved for F68, P85 and P106 were similar and did not allow a prioritisation of a specific formulation based on CMC alone. The concentrations of Pluronic (0.001 to 0.025%) used in the biological assays were based on the CMC values and were selected on the basis that they would be likely to consist of mainly unimers (0.001–0.025%); a mixture of unimers and micelles (0.1%) and mostly micelles (0.5%) respectively.

F68 micelles have a relatively small radius of 52.0 Å (Table 6). This attribute will increase stability, half-life and therefore circulation time of this Pluronic, since small micelles evade detection and destruction by the reticuloendothelial system. However, this small volume may also correlate to low drug loading (Table 5; Fig C in S1 File). In addition, the fact that pentamidine release from both F68 and P105 micelles is by diffusion would indicate that these Pluronics are unlikely to significantly prolong the circulation time of pentamidine (Fig D in S1 File).

Haemolysis of human red blood cells was not observed in the presence of 0.5%, 0.1%, 0.025%, 0.01%, and 0.001% P85, P105 or F68, the results being comparable to the negative control (0.05% DMSO). This suggests that an intravenous formulation containing P85, P105, or F68 would not lead to haemolysis at the tested concentrations, supporting the safety profile of Pluronic polymers for medical use[15, 78]. In agreement, no differences were reported in the terminal haematological values (including haemoglobin, packed cell volume, number of erythrocytes, total number of leukocytes) and blood-chemical values (including urea, total protein, alkaline phosphatase) obtained from rats who had received once daily intravenous doses of F68 (doses ranging from 10–1000 mg/kg body weight) or from rats who had been administered physiological saline for one month [79]. No morphological abnormalities were detected in the rats which received the 0–50 mg/kg daily dose of F68, however, rats which received the higher doses had detectable alterations i.e. the presence of foam cells in the lungs (dose was 500–1000 mg/kg) and focal cortical degenerative changes in the kidneys (dose was 100–1000 mg/kg).

Pentamidine caused a concentration-dependent inhibition of insulin secretion from MIN6 β-cells suggesting that this is one mechanism through which it could induce diabetes[9]. Pentamidine is known to be an agonist at imidazoline receptors [80], but it is unlikely that this explains its inhibitory effects on insulin secretion since β-cell imidazoline receptors are coupled to increased insulin release[81]. However, the imidazoline ligand idazoxan is reported to cause a concentration-dependent inhibition of β-cell viability[82], similar to the effects observed here with pentamidine, so it is possible that the reduction in insulin secretion is secondary to pentamidine-mediated activation of β-cell imidazoline receptors and impairment of cell viability. Pentamidine-induced diabetes is not thought to be reversible [9], and so testing for a marker of pancreatic off target adverse effects occurred early in the screening cascade. Importantly, a number of Pluronic formulations (P85, P105) were shown to increase the peripheral toxicity of pentamidine as measured by decreases in insulin secretion. In a human tissue cell model (HEK-293), P105 has previously been shown to cause dose dependent changes in cell viability[16]. However, a lead Pluronic (F68) was identified which demonstrated equivalent toxicity to unformulated pentamidine, on β-cell viability and insulin secretion. Supporting this formulation selection our studies also revealed that P85 and P105 at

0.01% and 0.5% concentrations caused loss of MDCK-MDR monolayer integrity, whereas F68 at concentrations up to 0.5% had no effect (Fig I in S1 File). A correlation between HLB and cytotoxicity has previously been observed with low cytotoxicity being guaranteed when the HLB of the polymer is ≥10 (Table 1)[30].

Importantly, all formulations tested did not prevent pentamidine killing *Trypanosoma brucei* blood stream form trypomastigotes. In fact, pure P85 and P105 were highly trypanocidal and F68-pentamidine formulations had a slight synergistic effect.

*In vitro* BBB studies indicated that there was an efflux process for pentamidine as also demonstrated in P-gp knockout mice studies [7]. However, we were unable to demonstrate an increase in pentamidine movement across the barrier in either direction, compared with unformulated pentamidine in any of our *in vitro* systems.

Further studies utilizing the *in situ* brain perfusion technique confirmed that the Pluronics (P85, P105 or F68) did not increase pentamidine delivery to the brain, including the choroid plexus, after either 10 or 30 minutes exposure. Our studies using *in situ* brain perfusions over 10 minutes in mice have shown that the P85, P105 and F68 formulations have a tendency to actually prevent uptake of pentamidine into brain tissue and/or vascular endothelial cells, which constitute an intact BBB. This may be related to interactions of the Pluronics with influx transporters for pentamidine (e.g. OCT1), although our *in vitro* BBB studies did not indicate that the pentamidine permeability was affected by the presence of F68, P85 and P105 (0.01% and 0.1%) in either direction. Importantly, a similar P85 induced reduction in BBB permeability was observed by other workers, [83] who noted a reduction in the rate of uptake into brain tissue of P85-leptin conjugates during the first 90 minutes after iv injection compared with native leptin. Despite this initial inhibition of P85-leptin influx, a greater overall concentration of the conjugate was measured in brain tissue after 4 hours, an observation that the authors ascribed to improved pharmacokinetic properties. Digoxin delivery to the brain has previously been determined 1–10 hr post-injection in mice and found to be significantly enhanced when Pluronic 85 is present [29].

Sucrose does not cross phospholipid membranes and was used in the brain perfusion experiments as a vascular space marker. An increase in [$^{14}$C(U)]sucrose would indicate that the integrity of the membrane or the tight junctions between cells had been compromised. Conversely, a decrease would suggest that the proportionate volume of tissue occupied by blood vessels had been reduced. It is therefore interesting that F68 has previously been shown to interact with the mechanisms that control vasoconstriction and vasodilation[84, 85] and could lead to the observed reduction in vascular space.

Interestingly, the *in vivo* mouse pharmacokinetic study revealed that the concentrations of pentamidine in brain parenchyma in this species seem high compared with data from human (using CSF rather than brain parenchyma) which indicated that less than 1% of the plasma pentamidine concentration is detected in CSF[86]. Furthermore, assessment of this lead formulation in an *in vivo* pharmacokinetic study confirmed that F68 did not increase pentamidine delivery to the brain under the conditions studied. This is not linked to partitioning of pentamidine inside the micelles as this is low, hence the use of Pluronic micelles to protect this drug after administration and extend its circulation time is probably limited. Although it may be related to the fact that F68 is hydrophilic and prefers to remain in the plasma than be distributed to organs [17].

Whilst there are limitations to all assay systems, the package of data generated by the team provided a compelling and robust data set. The screening cascade has successfully identified Pluronic-pentamidine formulations that harbour trypanocidal activity and do not increase the safety concerns centrally or peripherally (over unformulated pentamidine). However, the data suggested that we would not be able to significantly enhance brain exposure of pentamidine using the Pluronic (F68, P85 or P105) within a reasonable time frame and existing budget. We therefore drew the study to a close at milestone 2 (Fig 1). Importantly a significant body of

high-quality data has been generated as part of this project which may be highly relevant to other teams looking to understand block-copolymer architecture, further develop block-copolymers as nanocarriers, improve BBB penetration of drugs or to those looking to understand toxicity of pentamidine.

## Supporting information

**S1 File. Fig A—Pentamidine is returned to the blood from the capillary endothelial cell by P-gp and MRP. Pluronic P85 inhibits-mediated efflux (e.g. P-gp and MRP transport) by two mechanisms: the first through membrane fluidisation and the second through transient ATP depletion**. These effects are believed to be mediated by unimers (single polymer chains) [22, 20]. Inhibition of efflux should facilitate the accumulation of pentamidine in the human cerebral capillary endothelium and the murine choroid plexus epithelium, leading to higher concentrations of pentamidine. **Fig B—Pyrene fluorescence intensity dependence on pluronic concentration for F68, P85 and P105**. The CMC was determined using 18 different concentrations (range 0.0001 to 1 w/v%) of pure P85, P105 and F68. The value at each concentration is the mean of two samples, each prepared from a separate preparation of the stock solution. As expected, the curves show two inflection points. The first was taken as the CMC. **Fig C—Typical partition data for PTI fluorescence as a function of F68 and P105 concentration. Fig D—Drug release from dialysis cells measured over time. The experiments were conducted in water at 37˚C for concentrations as close as possible to *in vitro* conditions, within experimental limitations, namely, 1% w/v of Pluronics and 10mM PTI**. No significant differences between the Pluronics were observed and drug release is diffusion controlled (Fickian diffusion) under the experimental conditions. Pluronics micelles are not a barrier to drug release. **Fig E—SANS Pluronic data at 37˚C.** A) P85 5% B) F68 5% C) P85 5% / PTI 1% D) F68 5% / PTI 1% E) P85 5% / PTI 3% F) F68 5% / PTI 3%. **Fig F—The average number of Pluronic molecules found in a micelle ($N_{agg}$) and the number of micelles in our system (after they have equilibrated) ($N_{mic}$) as a function of the concentration of the F68 Pluronic in a system that contains F68 and 0.01 w/v% of L61 Pluronic**. In both plots, the black curve represents the results when considering both the L61 and F68 polymers in the mixture, and the blue dashed curve represents the data from the pure F68 simulated systems. In the top curve, the red curve represents the number of F68 in a micelle which contains both F68 and L61, and the green curve represents the number of L61 in a micelle. The results show that as we increase the concentration of F68, and therefore make the system more and more like the pure F68 system, the number of polymer molecules in a micelle and the number of micelles converge to that observed in the pure F68 system, as expected. Interestingly, it seems that from our simulations that L61 causes the aggregation of F68 to become slightly enhanced as the number of F68 in the average micelle is always larger than that found in the pure F68 micelles, which naturally results in their being fewer micelles. **Fig G—Effects of exposure of MIN6 β-cells to 0 (control), 1 or 100 μM pentamidine for 3 and 24 hours. Trypan blue uptake.** Blue staining demonstrates cells of compromised viability, highlighting the toxicity of 100 μM pentamidine to these cells after 3 hours exposure. **Fig H—Effects of exposure of MIN6 β-cells to 0, 1, 10 or 100 μM pentamidine and 0, 0.01, 0.025, 0.1 or 0.5% w/v% F68 for 24 hours**. Trypan blue uptake. Blue staining demonstrates cells of compromised viability, highlighting the toxicity of 100 μM pentamidine and 0.5% F68 to these cells. **Fig I—Apical to basolateral permeability of [$^{14}$C]sucrose in the presence of P85, P105, and F68 concentrations measured over 60 minutes**. Significant differences compared to control (no pluronic) was observed in the presence of P85 and P105 (***$p<0.001$, **$p<0.01$). All data are expressed as mean ± S.E.M, n = 3 wells. Data were analysed using one-way ANOVA with SigmaPlot 13.0. **Table A—Single**

point CNS side effect screening of pentamidine at a concentration of $1.0E^{-5}$ M (PERKIN ELMER study no. 13–9625). Details of the assay, reference $K_i$, reference compound and the radioligand/substrate used in the CNS side effects panel ligand binding assay are described. Values are expressed as the percent inhibition of specific binding and represent the average of duplicate tubes. Pentamidine could be described as active at that binding site if it showed inhibition of 50% or greater (see shaded boxes/compound hit true). Inhibition in the range of 20% to 49% indicated marginal activity at the receptor site and were not investigated further. The baseline range in these assays was considered -20% to +20% inhibition of binding activity. Compounds showing results in this range were considered inactive at this site. $K_i$ is the inhibitory constant and is reflective of the binding affinity of the drug for the receptor. **Table B—Inhibition of hKir2.1 potassium channel activity with pentamidine isethionate. Evaluated by the QPatch HT an automatic parallel patch clamp system.** The duration of exposure to each test concentration was 3 minutes. **Table C—A visual evaluation of the phase separation of Pluronics dispersions in pure water. Transparent is fully transparent. Opaque completely blocks light.** Slight indicates for slightly translucent (faintly white tint in the solution), and medium indicates obvious translucence. **Table D—A visual evaluation of the phase separation of Pluronic dispersions in saline. Transparent is fully transparent**. Opaque completely blocks light. Slight indicates for slightly translucent (faintly white tint in the solution), and medium indicates obvious translucence. **Table E—Stokes Radii of P105, P85 and F68 Micelles Obtained from DLS (1% w/w, 37˚C). Table F—The effect of P85, F68 and P105 on the apparent permeability of pentamidine isethionate across MDR1-MDCK cell monolayers in the basolateral to apical direction.** The apical to basolateral movement of pentamidine isethionate was below the limits of detection. The percentage recovery of pentamidine isethionate is also shown. Lucifer yellow permeation was below $0.5 \times 10^{-6}$ cm/s in all experiments confirming the integrity of the monolayer. Transcellular marker (propranolol) and Pgp and BCRP substrate (prazosin) apparent permeability values are also shown. **Table G—The effect of Pluronic P85 on the accumulation of [$^3$H(G)]pentamidine (15.7 nM) into brain tissues after 10 minutes of *in situ* perfusion.** All values have been corrected for vascular space by subtraction of the $R_{TISSUE}$% for [$^{14}$C(U)]sucrose from the $R_{TISSUE}$% for [$^3$H(G)]pentamidine. All values mean ± SEM. **Table H—The effect of Pluronic P105 on the accumulation of [$^3$H(G)]pentamidine (15.7 nM) into brain parenchyma after 10 minutes of *in situ* perfusion.** All values have been corrected for vascular space by subtraction of the $R_{TISSUE}$% for [$^{14}$C(U)] sucrose from the $R_{TISSUE}$% for [$^3$H(G)]pentamidine. **Table I—Accumulation of [$^3$H(G)]pentamidine (15.7 nM) after 10 minutes perfusion with or without pluronic F68 (not corrected for vascular space; Control A and 0.01% and 0.1% F68 experiments were carried out using MP Biomedicals dextran.** Control B and 0.5% F68 experiments were carried out using VWR dextran). **Table J—Accumulation of [$^{14}$C]sucrose after 10 minutes perfusion with or without Pluronic F68; Control A and 0.01% and 0.1% F68 experiments were carried out using MP Biomedicals dextran.** Control B and 0.5% F68 experiments were carried out using VWR dextran). **Table K—Accumulation of [$^3$H]pentamidine after 30 minutes perfusion with or without pluronic F68.** (Not corrected for vascular space). **Table L—Accumulation of [$^{14}$C]sucrose (B) after 30 minutes perfusion with or without pluronic F68.** (Not corrected for vascular space).
(DOCX)

## Acknowledgments

We are grateful for the project management support of Dr Gayle Chapman (Biomedical Catalyst Ltd), UK. We would like to acknowledge the support of Dr Surbi Gupta (King's IP & Licensing team) who was involved in the IP assessment of the project.

We would like to acknowledge the support of Drugs for Neglected Diseases initiative (DNDi) a collaborative, patients' needs-driven, non-profit drug research and development organization that is developing new treatments for Neglected Diseases.

We would like to thank Miss Raha Ahmadkhanbeigi (King's College London) for help with the haemolysis assay. We would like to thank Ms Anncharlott Berglar (King's College London) for help with acquisition of the preliminary data.

We would like to thank ISIS, Rutherford-Appleton Laboratory (Science and Technology Facilities Council, Didcot, Oxford) for the award of neutron beam time on the LOQ instrument to perform the SANS experiments.

According to the UK research councils' and Wellcome Trust Common Principles on Data Policy, all data supporting this study will be openly available as a supporting file.

## Author Contributions

**Conceptualization:** Mehmet Fidanboylu, Cécile A. Dreiss, Chris Lorenz, Mark Christie, Shanta J. Persaud, Vanessa Yardley, Simon L. Croft, Sarah A. Thomas.

**Data curation:** Cécile A. Dreiss, Chris Lorenz, Shanta J. Persaud, Sarah A. Thomas.

**Formal analysis:** Lisa Sanderson, Marcelo da Silva, Gayathri N. Sekhar, Rachel C. Brown, Hollie Burrell-Saward, Lea Ann Dailey, Cécile A. Dreiss, Chris Lorenz, Mark Christie, Shanta J. Persaud, Margarita Valero, Sarah A. Thomas.

**Funding acquisition:** Cécile A. Dreiss, Chris Lorenz, Mark Christie, Shanta J. Persaud, Vanessa Yardley, Simon L. Croft, Sarah A. Thomas.

**Investigation:** Lisa Sanderson, Marcelo da Silva, Gayathri N. Sekhar, Rachel C. Brown, Hollie Burrell-Saward, Bo Liu, Chris Lorenz, Mark Christie.

**Methodology:** Lea Ann Dailey, Cécile A. Dreiss, Chris Lorenz, Mark Christie, Shanta J. Persaud, Vanessa Yardley, Sarah A. Thomas.

**Project administration:** Sarah A. Thomas.

**Resources:** Lea Ann Dailey, Cécile A. Dreiss, Chris Lorenz, Shanta J. Persaud, Simon L. Croft, Sarah A. Thomas.

**Software:** Chris Lorenz.

**Supervision:** Sarah A. Thomas.

**Validation:** Sarah A. Thomas.

**Visualization:** Sarah A. Thomas.

**Writing – original draft:** Lisa Sanderson, Marcelo da Silva, Gayathri N. Sekhar, Hollie Burrell-Saward, Cécile A. Dreiss, Chris Lorenz, Mark Christie, Shanta J. Persaud, Sarah A. Thomas.

**Writing – review & editing:** Marcelo da Silva, Lea Ann Dailey, Cécile A. Dreiss, Chris Lorenz, Shanta J. Persaud, Simon L. Croft, Sarah A. Thomas.

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
