## [Decision Letter · Decision Letter 0]

19 Aug 2020

Dear Dr Thomas,

Thank you very much for submitting your manuscript "Drug reformulation for a
neglected disease.  The NANOHAT project to develop a safer more effective sleeping
sickness drug." for consideration at PLOS Neglected Tropical Diseases. As with all
papers reviewed by the journal, your manuscript was reviewed by members of the
editorial board and by several independent reviewers. In light of the reviews (below
this email), we would like to invite the resubmission of a significantly-revised
version that takes into account the reviewers' comments. 

We recognise that you have as a team performed a thorough analyses of this approach
to drug treatment, which has failed. We consider this important for the community to
know. PLoS NTD readership will be lost with the majority of the data in this paper.
We asked one reviewer versed in drug pharmacokinetics to give us suggestions for
improving readership and the other versed in the blood brain barrier.

We ask that you take their hard work to heart and move many of the details to
supplemental and make the main data accessible to our readership in a more broad
fashion. PLoS is committed to publishing all data whether negative or not. We hope
that you take our suggestions seriously.

Forgive us for taking so long, these times of COVID work in mysterious ways.

With all good wished Jayne

We cannot make any decision about publication until we have seen the revised
manuscript and your response to the reviewers' comments. Your revised manuscript is
also likely to be sent to reviewers for further evaluation.

Sincerely,

Jayne Raper, PhD

Associate Editor

Ana Rodriguez

Deputy Editor

We recognise that you have as a team performed a thorough analyses of this approach
to drug treatment, which has failed. We consider this important for the community to
know. PLoS NTD readership will be lost with the majority of the data in this paper.
We asked one reviewer versed in drug pharmacokinetics to give us suggestions for
improving readership and the other versed in the blood brain barrier.

We ask that you take their hard work to heart and move many of the details to
supplemental and make the main data accessible to our readership in a more broad
fashion. PLoS is committed to publishing all data whether negative or not. We hope
that you take our suggestions seriously.

Forgive us for taking so long, these times of COVID work in mysterious ways.

With all good wished Jayne

Reviewer's Responses to Questions

**Key Review Criteria Required for Acceptance?**

**Methods**

-Are the objectives of the study clearly articulated with a clear testable hypothesis
stated?

-Is the study design appropriate to address the stated objectives?

-Is the population clearly described and appropriate for the hypothesis being
tested?

-Is the sample size sufficient to ensure adequate power to address the hypothesis
being tested?

-Were correct statistical analysis used to support conclusions?

-Are there concerns about ethical or regulatory requirements being met?

Reviewer #1: The objectives were clearly stated, and study design was clearly
stated.

Reviewer #2: (No Response)

**Results**

-Does the analysis presented match the analysis plan?

-Are the results clearly and completely presented?

-Are the figures (Tables, Images) of sufficient quality for clarity?

Reviewer #1: The analysis is consistent with the plan, and completely presented;
perhaps TOO completely. See below.

Reviewer #2: (No Response)

**Conclusions**

-Are the conclusions supported by the data presented?

-Are the limitations of analysis clearly described?

-Do the authors discuss how these data can be helpful to advance our understanding of
the topic under study?

-Is public health relevance addressed?

Reviewer #1: (No Response)

Reviewer #2: (No Response)

**Editorial and Data Presentation Modifications?**

Reviewer #1: (No Response)

Reviewer #2: (No Response)

**Summary and General Comments**

Reviewer #1: This is a very (very) thorough article that describes the studies aimed
at understanding the effect of formulation on CNS activity for pentamidine. The
questions were: (1) does the formulation facilitate transmission to the CNS; (2)
does formulation affect toxicity or efficacy of pentamidine (and is the formulation
itself toxic)? Further experiments were performed in order to fully characterize the
formulations.

The outcome of the experiments was less than successful, given the goal for CNS
activity of pentamidine. Nonetheless, there is a lot of very useful information
here, and the thoroughness of the work affords a great deal of useful information
and rigorous data.

That said, this article is much, much too long. The authors would benefit from more
careful selection of what experimental designs, details, and results belong in the
main body of the paper, and what could be most usefully put into the supporting
information. I believe that if this paper were to be published with this level of
depth and detail, it should be in a pharmaceutics journal, rather than PLOS NTDs. To
the PLOS NTD readership, I believe that this paper is too unfocused, and that the
salient experimental results and interpretation should be severely streamlined.

Some specific comments:

1. There are a good number of grammatical errors; the paper needs a careful
re-read.

2. Line 143 - how does the successful use of AmpB micelles indicate that pluronics
"are also active agents with key biological functions"?

3. Line 151. I think that if there's eivdence that pluronics improve BBB penetration
it would be good to reference that here.

4. The Figure 1 content is very dense, the fonts are very small, and the color coding
is not explained. In addition, the "SAR feedback loop" is never addressed in the
paper -- how would the results feed back into the in silico and subsequent
experiments? And I don't believe it's comment for the term "structure activity
relationships" to be used referring to formulations...

5. Line 396 and following -- the last sentence of this paragraph is confusing....were
only two healthy volunteers' blood used? If so, maybe just say this in the first
sentence.

6. Line 471 - what is the difference between "permeability' and "accumulation"
formats?

7. In the animal studies, why were opposite sexes used in the two different mouse
strains? 

8. Table 3 - the last three entries show estimated Ki values. How were these
estimated? And all three are estimated to be exactly 10 uM?

9. Table 4. Please consider whether four significant figures are appropriate
here.

10. Line 824 and following. I'm surprised to see that pluronics are so active in
inhibiting insulin secretion...and/or that this wouldn't already have been a known
phenomenon given the formulations' prevelance. IF it is know, this should be stated
and referenced. 

11. Line 1061. I'm not sure that I saw that any MD simulations were performed, or how
they were utilized (they don't seem to be mentioned anywhere else.

Reviewer #2: See attached comments called "Reviewer #2 Comments" or the file named
"Reviewer2_PNTD-D-20-00526"

PLOS authors have the option to publish the peer review history of their article
(what does this mean?). If published, this will
include your full peer review and any attached files.

If you choose “no”, your identity will remain anonymous but your review may still be
made public.

**Do you want your identity to be public for this peer review?** For
information about this choice, including consent withdrawal, please see our
Privacy Policy.

Reviewer #1: No

Reviewer #2: No
---

## [Decision Letter · Decision Letter 1]

26 Feb 2021

Dear Dr Thomas,

We are pleased to inform you that your manuscript 'Drug reformulation for a neglected
disease.  The NANOHAT project to develop a safer more effective sleeping sickness
drug.' has been provisionally accepted for publication in PLOS Neglected Tropical
Diseases.

Best regards,

Ana Rodriguez

Deputy Editor

Ana Rodriguez

Deputy Editor

Reviewer's Responses to Questions

**Key Review Criteria Required for Acceptance?**

**Methods**

-Are the objectives of the study clearly articulated with a clear testable hypothesis
stated?

-Is the study design appropriate to address the stated objectives?

-Is the population clearly described and appropriate for the hypothesis being
tested?

-Is the sample size sufficient to ensure adequate power to address the hypothesis
being tested?

-Were correct statistical analysis used to support conclusions?

-Are there concerns about ethical or regulatory requirements being met?

Reviewer #2: See comment to Editor

**Results**

-Does the analysis presented match the analysis plan?

-Are the results clearly and completely presented?

-Are the figures (Tables, Images) of sufficient quality for clarity?

Reviewer #2: See comment to Editor

**Conclusions**

-Are the conclusions supported by the data presented?

-Are the limitations of analysis clearly described?

-Do the authors discuss how these data can be helpful to advance our understanding of
the topic under study?

-Is public health relevance addressed?

Reviewer #2: See comment to Editor

**Editorial and Data Presentation Modifications?**

Reviewer #2: See comment to Editor

**Summary and General Comments**

Reviewer #2: See comment to Editor

PLOS authors have the option to publish the peer review history of their article
(what does this mean?). If published, this will
include your full peer review and any attached files.

If you choose “no”, your identity will remain anonymous but your review may still be
made public.

**Do you want your identity to be public for this peer review?** For
information about this choice, including consent withdrawal, please see our
Privacy Policy.

Reviewer #2: No

---

## [Editor Report · Acceptance letter]

1 Apr 2021

Dear Dr Thomas,

We are delighted to inform you that your manuscript, "Drug reformulation for a
neglected disease. The NANOHAT project to develop a safer more effective sleeping
sickness drug.," has been formally accepted for publication in PLOS Neglected
Tropical Diseases.

Best regards,

Shaden Kamhawi

co-Editor-in-Chief

Paul Brindley

co-Editor-in-Chief
